# A robust activity marking system for exploring active neuronal ensembles

**Andreas T Sørensen[1,†,‡], Yonatan A Cooper[1,†,§], Michael V Baratta[2], Feng-Ju Weng[1], Yuxiang Zhang[1], Kartik Ramamoorthi[1], Robin Fropf[3,4,5], Emily LaVerriere[1], Jian Xue[1], Andrew Young[1], Colleen Schneider[1], Casper René Gøtzsche[1], Martin Hemberg[6], Jerry CP Yin[3,4], Steven F Maier[7], Yingxi Lin[1]***

[1]McGovern Institute for Brain Research, Department of Brain and Cognitive Sciences, Massachusetts Institute of Technology, Cambridge, United States; [2]Institute for Behavioral Genetics, University of Colorado Boulder, Boulder, United States; [3]Department of Genetics, University of Wisconsin-Madison, Madison, United States; [4]Department of Neurology, University of Wisconsin-Madison, Madison, United States; [5]Neuroscience Training Program, University of Wisconsin-Madison, Madison, United States; [6]Wellcome Trust Sanger Institute, Hinxton, United Kingdom; [7]Department of Psychology and Neuroscience, University of Colorado Boulder, Boulder, United States

**\*For correspondence:** yingxi@mit. edu

[†]These authors contributed equally to this work

**Present address:** [‡]Department of Neuroscience and Pharmacology, University of Copenhagen, Copenhagen, Denmark; [§]David Geffen School of Medicine, University of California Los Angeles, Los Angeles, United States

**Competing interests:** The authors declare that no competing interests exist.

**Abstract** Understanding how the brain captures transient experience and converts it into long lasting changes in neural circuits requires the identification and investigation of the specific ensembles of neurons that are responsible for the encoding of each experience. We have developed a Robust Activity Marking (RAM) system that allows for the identification and interrogation of ensembles of neurons. The RAM system provides unprecedented high sensitivity and selectivity through the use of an optimized synthetic activity-regulated promoter that is strongly induced by neuronal activity and a modified Tet-Off system that achieves improved temporal control. Due to its compact design, RAM can be packaged into a single adeno-associated virus (AAV), providing great versatility and ease of use, including application to mice, rats, flies, and potentially many other species. Cre-dependent RAM, CRAM, allows for the study of active ensembles of a specific cell type and anatomical connectivity, further expanding the RAM system's versatility.

## Introduction

Neurons form ensembles that encode experiences. This has been demonstrated in the past several decades by in vivo electrophysiological and calcium imaging experiments in which the activity of neuronal ensembles has been correlated with behavior in active animals (*Buzsáki, 2004*; *Grewe and Helmchen, 2009*). Understanding the process whereby experience is converted to long-term memory and consequent behavioral modification requires that *relevant* ensembles of neurons be defined precisely and genetically to allow functional interrogation and manipulation.

Transcription events triggered within neurons by neuronal activity are key to neural circuit plasticity, ensemble formation, and ultimately information storage (*Alberini, 2009*; *Cole et al., 1989*; *Guzowski et al., 2001*; *Johansen et al., 2011*). Experience-dependent transcription events thus present a promising way to genetically identify neurons responsible for encoding learned experiences in vivo. However, the transcriptional profile must fit the following two criteria: (1) very low basal expression in the absence of salient experience and (2) strong induction by neuronal activity

**eLife digest** Every experience – be it a sight, a sound or a memorable event – activates a unique set of neurons within the brain that together are known as a neuronal ensemble. Identifying these ensembles is key to deciphering how the brain represents experiences and stores them in memory. The most commonly used method for doing so at present relies upon a class of genes called immediate early genes (or IEGs for short). Whenever a neuron becomes active, it switches on its IEGs. By genetically modifying animals to use this mechanism to drive the production of protein markers – such as a fluorescent protein – it is possible to visualize and control the neurons that become activated in response to a stimulus.

However, existing IEG-based systems for detecting neuronal activity are not ideal. In particular, these systems could be made more sensitive (so that they are more likely to respond to neuronal activity) and more specific (so that they are more likely to respond *only* to relevant neuronal activity). Sørensen, Cooper et al. have now developed a new system for tagging recently activated neurons that offers a number of advantages over its predecessors.

Known as Robust Activity Marking (RAM), the new system consists of a specially designed DNA sequence that is switched on by neuronal activity. Compared with currently existing systems, the RAM system has low levels of background activity, meaning that it only becomes active in actively firing neurons. It is also extremely sensitive and gives a robust signal. An additional advantage of the RAM system is that the timing of its activation can be precisely controlled. This is useful for identifying those neurons that become active in response to one particular sensory stimulus.

The DNA elements in the RAM system that respond to neuronal activity are conserved, which means it could be used in a variety of species, from fruit flies to primates. The relatively small size of the RAM system means that, in contrast to other IEG-based systems, it can be introduced into brains by packaging the entire DNA sequence inside a virus particle that can infect a wide range of experimental species. Finally, the design of the RAM system allows it to be targeted to specific subtypes of neurons and to cells that are connected in particular ways.

Together, the multiple advantages of the RAM system over traditional IEG-based systems should make it possible for neuroscientists from many different fields to explore how the brain stores experiences in patterns of neuronal activity.

associated with experience and behavior. Immediate early genes (IEGs) such as *Fos, Arc* and *Egr1* meet these criteria quite well (*Guzowski et al., 2001*), and their promoters have been used to control the expression of effector genes such as fluorescent proteins and opsins in genetically engineered mouse lines, allowing active ensemble labeling and functional perturbation, respectively (*Barth et al., 2004*; *Guenthner et al., 2013*; *Koya et al., 2009*; *Reijmers et al., 2007*; *Smeyne et al., 1992*; *Wang et al., 2006*; *Eguchi and Yamaguchi, 2009*; *Denny et al., 2014*). However, significant technical obstacles greatly limit the usability of these systems.

The biggest challenge is to improve the sensitivity and selectivity of neuronal ensemble identification. Existing systems suffer from high background, i.e. labeling of neurons unrelated to the experimental stimulus of interest, which confounds precise identification of the relevant active ensemble. The level of background labeling is determined by the characteristics of the IEG promoter used and the method of temporal control used to isolate events happening within a desired experimental time window. Therefore, to address the problem of background labeling, we wanted to develop an IEG-sensitive promoter with an optimized activity-dependent induction profile and incorporate it into a platform with improved temporal control of effector gene expression.

In addition, the use of transgenic reporter lines in existing systems requires laborious breeding and is experimentally inflexible. Therefore, we also aimed to develop a system in which both the activity-dependent transcription component and the effector genes for neural circuit interrogation are delivered using a single adeno-associated virus (AAV). In addition to bypassing the requirement for multiple transgenic mouse lines, an entirely viral system can also be used in species other than the mouse.

Here we present a virus-based platform for the analysis of active neuronal ensembles, which we call the Robust Activity Marking (RAM) system. The following features of the RAM system make it highly selective, sensitive, and versatile: (1) a synthetic neuronal activity-dependent promoter with very low expression in basal conditions prior to a designated experience and strong induction by neural activity during the experience for robust ensemble labeling; (2) a modified Tet-Off system that provides improved temporal control; (3) small size, well within the packaging limit of a single AAV; (4) modular design so that the promoter and effector genes can be easily substituted to address different experimental questions; and (5) proven transferability to species other than the mouse, making it a valuable tool for the wider neuroscience community. We demonstrate the use of the RAM system to interrogate active neuronal ensembles in several different regions of the murine and drosophila brain.

## Results

### Design of a synthetic activity-regulated promoter

To find a small promoter broadly responsive to neuronal activity, we searched for highly enriched DNA elements among 11,830 previously identified neuronal activity-regulated enhancers (*Kim et al., 2010*). The Activator Protein 1 (AP-1) site (TGANTCA), a consensus sequence for the FOS/JUN family transcription factors, was initially identified as the most highly enriched motif (*Figure 1—figure supplement 1*). We further considered that multiple transcription factors often bind enhancers cooperatively or competitively to tightly regulate gene expression in a combinatorial manner (*Hermsen et al., 2006*). We therefore tried combining the AP-1 site with the binding motif of the neuronal-specific activity-dependent gene *Npas4* (*Ramamoorthi et al., 2011*) (NRE: TCGTG) to increase the sensitivity and specificity of our promoter.

We inserted the core NRE/AP-1 DNA motifs into the central midline element (CME), a characterized transcriptional regulatory sequence whose secondary structure is favorable for transcription activation (*Wharton et al., 1994*), resulting in a 24 bp 'enhancer module.' Unless otherwise stated, we assembled promoters by placing four tandem repeats of an enhancer module upstream of the human *FOS* minimal promoter, resulting in a 199 bp synthetic promoter (*Figure 1a*). We assayed both the absolute transcriptional strength, reflected by the relative luciferase value, and the activity dependence and sensitivity, represented by the fold induction between stimulated and unstimulated conditions, of the synthetic promoters using dual luciferase assay in cultured neurons (See 'Methods' for details).

The relative position of the NRE and AP-1 sites turned out to be critical. When these motifs partially overlapped by two base-pairs to form a 10mer TCG**TG**ANTCA, the resulting synthetic promoter had the strongest activity-dependent induction profile (*Figure 1—figure supplement 2a*). We named this promoter $P_{RAM}$. The CME alone was not responsive to activity (*Figure 1a*). Furthermore, $P_{RAM}$ performance was not improved by replacing the NRE/AP-1 element with other highly enriched AP-1 containing 10mers found in the 11,830 neuronal activity-regulated enhancers, indicating the significance of the NRE and AP-1 combination in our design (*Figure 1a*, *Figure 1—figure supplement 2b*). In general, we found no correlation between 10mer enrichment and fold induction in the luciferase assay (*Figure 1a*, *Supplementary file 1*).

We proceeded with further characterization of $P_{RAM}$ and found that removing or substituting the minimal promoter (*Figure 1b*; *Figure 1—figure supplement 2c*), or changing the number of enhancer modules substantially reduced the fold induction (*Figure 1—figure supplement 2d*). When compared to several other activity-regulated promoters, $P_{RAM}$ had among the highest fold induction (*Figure 1c*) and the highest relative luciferase activity in response to neuronal stimulation (*Figure 1—figure supplement 3a*). We compared $P_{RAM}$ to another recently developed synthetic activity-dependent promoter, ESARE, which was based on the promoter of the IEG *Arc* (*Kawashima et al., 2013*). ESARE had a significantly lower fold induction, due to its high basal activity (*Figure 1d*). Notably, $P_{RAM}$ has by far the largest fold induction of any activity-dependent promoter small enough to be packaged into an AAV.

We next verified that $P_{RAM}$ has the characteristics expected from the presence of the FOS and NPAS4 binding motifs. Indeed, over-expression of FOS, NPAS4, or both proteins in combination was sufficient to drive transcription through $P_{RAM}$ (*Figure 1e*). Expression from $P_{RAM}$ was strongly

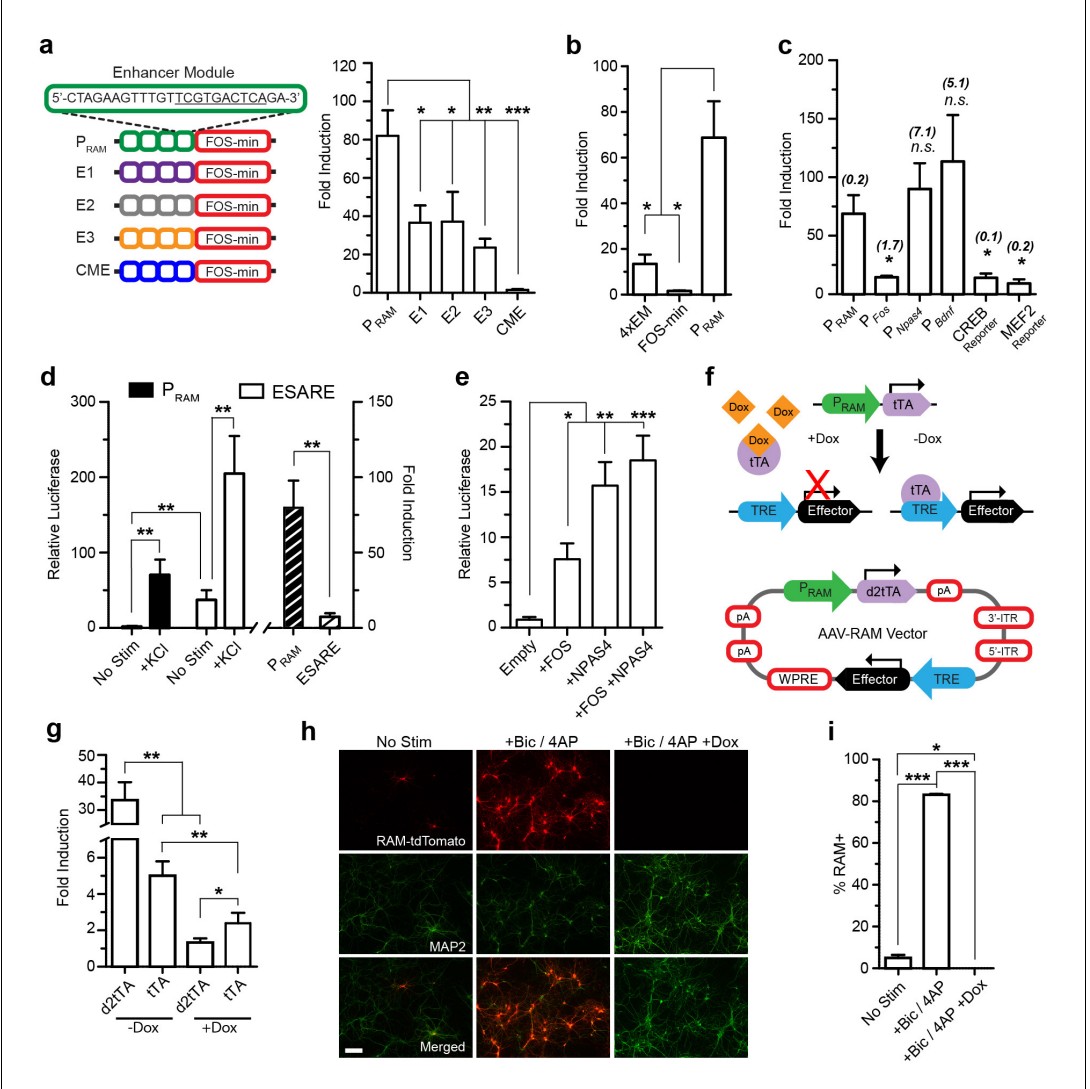

**Figure 1.** Design and characterization of the RAM promoter ($P_{RAM}$) and AAV-based RAM system in vitro. Unless indicated otherwise, cultured mouse hippocampal neurons were transfected with luciferase reporter constructs on DIV5 and stimulated with 35 mM KCl for 6 hr on DIV 7 or 8 (See *Supplementary file 2* for details). The relative luciferase value is the absolute luciferase value normalized against the internal control. The fold induction is the ratio of the relative luciferase values in stimulated and unstimulated conditions. (a) $P_{RAM}$ has higher fold induction than promoters in which the NRE/AP-1 element is replaced by other elements found enriched in the 11,830 activity-regulated enhancers (See 'Methods'). CME is not regulated by activity. Each construct contains four enhancer modules (EM) inserted upstream of the *FOS* minimal promoter (left). The relative luciferase values are shown in *Figure 1—figure supplement 2b*. n = 5–7 separate experiments per condition, one-way ANOVA, Tukey's post-hoc test. (b) The robust activity response of $P_{RAM}$ is a result of the combination of the four repeated RAM EM and *FOS* minimal promoter, which alone are respectively weakly responsive or non-responsive to neuronal activity. n = 5–10 separate experiments per condition, one-way ANOVA, Tukey's post-hoc test. (c) Comparison of $P_{RAM}$ to various activity-dependent promoters. Promoter size (Kb) is shown in brackets. The relative luciferase values are shown in *Figure 1—figure supplement 3a*. n = 4–10 separate experiments per condition, Student's t-test. (d) Comparison of $P_{RAM}$ and ESARE. n = 6–8 separate experiments per condition, Student's t-test. (e) $P_{RAM}$ can be driven by overexpression of FOS and NPAS4 individually or in combination. n = 9–10 separate experiments per condition, one-way ANOVA, Dunnett's post-hoc test. (f) Top, schematic outline of Tet-OFF system with $P_{RAM}$ driving tTA expression. Binding of tTA protein to the TRE promoter is prevented by Dox administration (+Dox); withdrawal of Dox (-Dox) allows downstream effector gene transcription. Bottom, schematic diagram of the AAV-RAM construct with critical genetic elements outlined. (g) Comparison of $P_{RAM}$-d2tTA and $P_{RAM}$-tTA by assaying pTRE-luciferase activity with and without Dox (+Dox, −Dox). The relative luciferase values are shown in *Figure 1—figure supplement 3e*. n = 3 separate experiments per condition, Student's t-test. (h) Representative images of hippocampal neurons grown on a glia monolayer and infected with AAV-RAM-tdTomato (red). Neurons are identified by MAP2 staining (green). Cultures were either left undisturbed (No Stim) or stimulated with bicuculline and 4AP (+Bic/4AP), either with (+Dox) or without (-Dox) doxycycline added. The scale bar is 150 μm and applies to all images. (i) Quantification of **h**. Percentages of neurons (MAP2+) that are RAM+ (%RAM+) are plotted. n = 3 separate experiments per condition, one-way ANOVA, Tukey's post-hoc test. All data in **a–e**, **g** and **i** are mean ± SEM. *p<0.05, **p<0.01, ***p<0.001.

*Figure 1 continued on next page*

*Figure 1 continued*

The following figure supplements are available for figure 1:

**Figure supplement 1.** Position-weight matrix of the top-ranking motif (AP-1, TGANTCA) identified by a Weeder de novo motif search.
**Figure supplement 2.** Characterization of enhancers by luciferase assay.
**Figure supplement 3.** Characterization of the $P_{RAM}$ promoter in vitro.
**Figure supplement 4.** Quantification of AAV-RAM-tdTomato infection of neuron-glia co-cultures determined by immunocytochemistry.

induced by $Ca^{2+}$ influx (*Figure 1—figure supplement 3b*) and by various growth factors known to induce *Fos* expression (*Lin et al., 2008*) (*Figure 1—figure supplement 3c*). When expressed in glia, $P_{RAM}$ had minimal basal activity and no activity-dependent induction (*Figure 1—figure supplement 3d*). We concluded that $P_{RAM}$ is a neuron-specific and highly activity-dependent promoter.

## The RAM system: a modular bicistronic AAV vector with optimized temporal control of gene expression

We next constructed the RAM system, incorporating $P_{RAM}$ into the doxycycline-dependent Tet-Off system (*Gossen et al., 1995*). Specifically, $P_{RAM}$ drives the expression of the tetracycline transactivator (tTA), which efficiently activates the tTA-responsive element (TRE) promoter driving the effector gene cassette. Binding of tTA to TRE can be blocked by administration of the antibiotic doxycycline (Dox), preventing expression of the effector gene (*Figure 1f*) and allowing temporal control of gene expression.

Our preliminary evaluation of the conventional Tet-Off system showed that the accumulation of tTA outside of the designated experimental window contributed to the undesirable background expression of effector genes. We therefore created a destabilized version of tTA, d2tTA, by fusing the degradation domain of mouse ornithine decarboxylase (MODC) to the N-terminus of tTA. Using a luciferase assay after co-transfecting a TRE-luciferase plasmid with either $P_{RAM}$-tTA or $P_{RAM}$-d2tTA into cultured neurons, we found that using d2tTA led to significantly lower basal expression (*Figure 1—figure supplement 3e*), tighter Dox regulation (*Figure 1g*), and improved fold induction by a factor of 7–8 compared with conventional tTA (*Figure 1g*). *Figure 1f* shows the final design of the RAM system AAV vector. Due to the compact design and small size of $P_{RAM}$, all the necessary components fit within the packaging limit of a single AAV vector (4.9 Kb) with room for an effector gene of up to 1.8 Kb in size.

We validated the RAM system first in cultured neurons, using the red fluorophore tdTomato (tdT) as the effector gene. Using an optimized titer of AAV-RAM-tdT (*Figure 1—figure supplement 4a*), close to 100% of cultured mouse hippocampal neurons were infected at DIV7. With Dox absent throughout, at DIV14 approximately 5% of neurons (MAP2-positive cells) were labeled with tdT under unstimulated conditions, while nearly 90% of neurons were labeled following bicuculline (Bic) and 4-aminopyridine (4AP) stimulation (*Figure 1h–i*). RAM labeling was tightly controlled by Dox, with no labeling cells detected when Dox was present after infection (*Figure 1h–i*). No glial cells were tdT positive (*Figure 1—figure supplement 4b*). Thus, our AAV-based RAM system showed strong induction and low background, as desired.

## Optimizing in vivo labeling using the RAM system

We carried out a series of experiments to determine optimal working parameters for our AAV-based RAM system in vivo. For these experiments, we chose the nuclear localized, red-shifted fluorophore NLS-mKate2 as the effector gene. For all in vivo experiments in mice, AAV-Ef1α-EGFP, which constitutively expresses EGFP in infected neurons, was co-injected as an infection control. The titer of the EGFP control virus was adjusted so that nearly 100% of neurons at the injection site were infected. To optimize the RAM virus, animals were placed on a Dox diet 24 hr before viral injection into the hippocampus, and allowed to recover for at least 7 days before administration of kainic acid to induce

seizures, which robustly stimulates the limbic system and fully activated $P_{RAM}$ in the majority of AAV-RAM infected neurons. With an optimized titer, more than 90% of EGFP+ neurons were also RAM+ after seizure. Using this setup, we found that animals should optimally be off Dox for 24–48 hr before behavioral manipulation (in this case the manipulation was seizure, *Figure 2a–d*), and that effector gene expression peaked within 24 hr following behavioral manipulation (*Figure 2a,e–g*). In addition, a further expression of the effector gene was completely blocked 24 hr after the animals were placed back on Dox chow (*Figure 2—figure supplement 1a–d*).

## Labeling active ensembles in the hippocampus

We next tested the ability of RAM to label neuronal ensembles activated by salient experience, using contextual fear conditioning (CFC) as a model behavioral paradigm. Animals were kept on Dox chow for at least 1 week after viral injection, switched to Dox-free chow 48 hr before CFC, and sacrificed 24 hr after CFC (*Figure 3a*). First we virally targeted the dentate gyrus (DG) sub-region (*Figure 3b*), which is known to express IEGs after CFC (*Liu et al., 2012*). Each cohort of animals was divided into the following five treatment groups: (1) left unperturbed in the home cage (HC), (2) CFC, (3) exposed to the novel context only without shock (Context Only) and (4) exposed to immediate shocks (Shock Only) and (5) kainic acid-induced seizure (*Figure 3c–d*). We quantified the percentage of EGFP+ cells that were RAM+ (labeled with NLS-mKate2) in each treatment group. We found a significant increase in RAM+ cells in both the CFC and Context Only group, compared to the HC animals (*Figure 3c*). Importantly, RAM labeled a very similar number of DG neurons in animals exposed to Context Only as in those that underwent CFC (*Figure 3c*), which is consistent with published studies showing that the hippocampus forms contextual representations independent of shock delivery (*Rudy and O'Reilly, 1999*; *Fanselow, 2000*) and that similar levels of IEG induction occur for these two conditions (*Ramamoorthi et al., 2011*). Furthermore, the Shock Only condition did not induce RAM activation at all (*Figure 3c*), suggesting that RAM labeling was specific to neuronal ensembles involved in contextual learning. Interestingly, such learning-specific induction is one of the unique properties of *Npas4* (*Ramamoorthi et al., 2011*; *Sun and Lin, 2016*). These results suggest that the RAM system labels an active population of cells in an experience-dependent manner. Importantly, mice that were kept on Dox chow throughout CFC treatment showed a complete lack of RAM+ cells in the DG (*Figure 3—figure supplement 1a–b*).

We repeated these experiments in the CA3 sub-region of the hippocampus and saw robust activity-dependent induction of RAM labeling after CFC along with low HC expression, consistent with a previous report of IEG expression in this region (*Guenthner et al., 2013*) (*Figure 3—figure supplement 2a–d*). Interestingly, CFC treatment resulted in RAM+ labeling of 11.4% of infected CA3 pyramidal neurons, compared with 4.4% of infected DG granule cells, similar to published percentages of cells labeled by IEGs in response to CFC in these regions (*Kubik et al., 2007*; *Leutgeb et al., 2007*).

To determine how long expression of the effector gene persists, animals were placed back on Dox 24 hr following CFC, and mKate2 labeling was measured in the following weeks. Labeling persisted for at least 1 week with no noticeable signal loss, began to decay at 2 weeks and was undetectable after 4 weeks (*Figure 3—figure supplement 3a–d*). To test whether the RAM system drives effector gene expression sufficiently for functional manipulation the following day, we injected AAV-RAM-ChR2 or AAV-RAM-ArchT, which expressed activating channelrhodopsin and inhibiting archeorhodopsin, respectively, into the DG. Ex vivo slice recordings of sparsely RAM+ labeled dentate granule cells 24 hr after CFC revealed that active ensembles could be activated or inhibited by light (*Figure 3—figure supplement 4a–j*), paving the way for future functional manipulation of neuronal ensembles labeled by the RAM system.

## Validating functional relevance of RAM+ ensembles

We next designed experiments to test whether the active ensembles of neurons labeled by RAM are tightly associated with specific experiences. Because ensembles activated by different contexts can be distinguished within the DG region (*Liu et al., 2012*; *Garner et al., 2012*), and ensembles activated by a particular context are preferentially re-activated following re-exposure to the same context (*Barth et al., 2004*; *Reijmers et al., 2007*; *Liu et al., 2012*; *Cruz et al., 2013*;

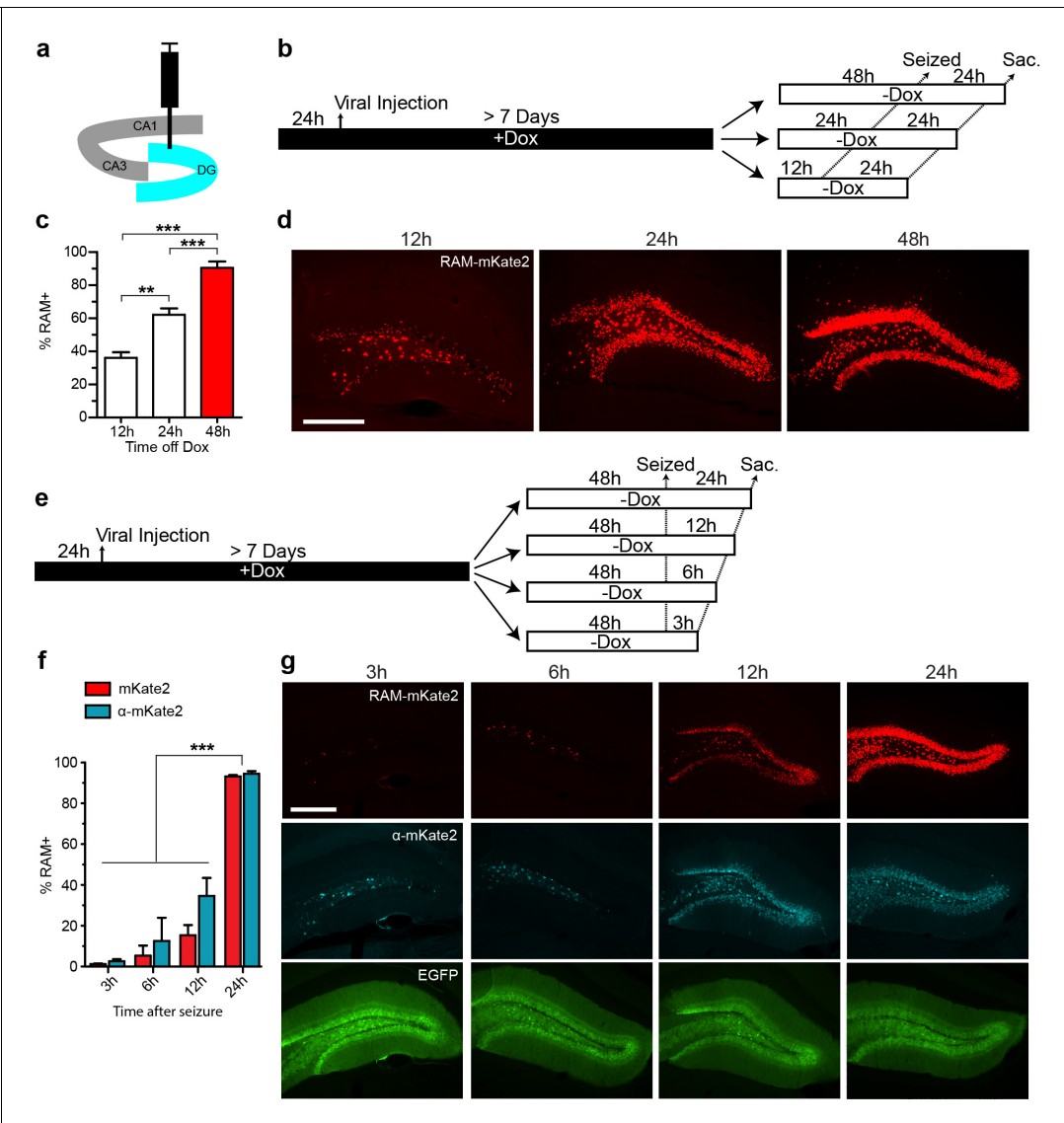

**Figure 2.** Optimizing the working parameters for the AAV-RAM system in vivo. (a) Schematic of the hippocampus showing the CA1, CA3 and DG regions with the injection site (DG) highlighted in blue. (b–d) Experiments to determine the time required for Dox clearance to allow maximal effector gene expression. (b) Experimental scheme. Animals were co-infected with AAV-RAM-NLS-mKate2 and AAV-Ef1α-EGFP and kept on Dox diet (+Dox) for a minimum of 7 days. The animals were taken off Dox diet (−Dox) 12, 24, or 48 hr before kainic acid treatment to induce seizures and sacrificed 24 hr after this treatment. (c) Percentage of RAM+ cells among total EGFP+ cells in the DG stratum granulosum when seized 12, 24 or 48 hr after Dox removal. n = 3–4 animals per group, one-way ANOVA, Tukey's post-hoc test. (d) Representative images from the data quantified in c. The scale bar is 150 μm and applies to all images. (e–g) Experiments to determine the maturation of effector gene expression. (e) Experimental scheme. Animals were treated similarly to a, and 48 hr following Dox removal (-Dox), animals were seized then sacrificed 3, 6, 12, or 24 hr later. (f) Percentage of RAM+ cells among total EGFP+ cells in the DG stratum granulosum 3, 6, 12, or 24 hr after seizure. RAM+ cells are detected by fluorescence (mKate2, red) or immuno-staining (α-mKate2, cyan). n = 2–4 animals per group, two-way ANOVA, Bonferroni post-hoc test. (g) Representative images from the data quantified in f. Upper row: AAV-mKate2 (red), middle row: mKate2 detected by antibody (α-mKate2, cyan), lower row: EGFP (green). The scale bar is 150 μm and applies to all images. Data in c and f are mean ± SEM. **p<0.01, ***p<0.001.

The following figure supplement is available for figure 2:

**Figure supplement 1.** Rapid suppression of RAM expression on returning to Dox diet.

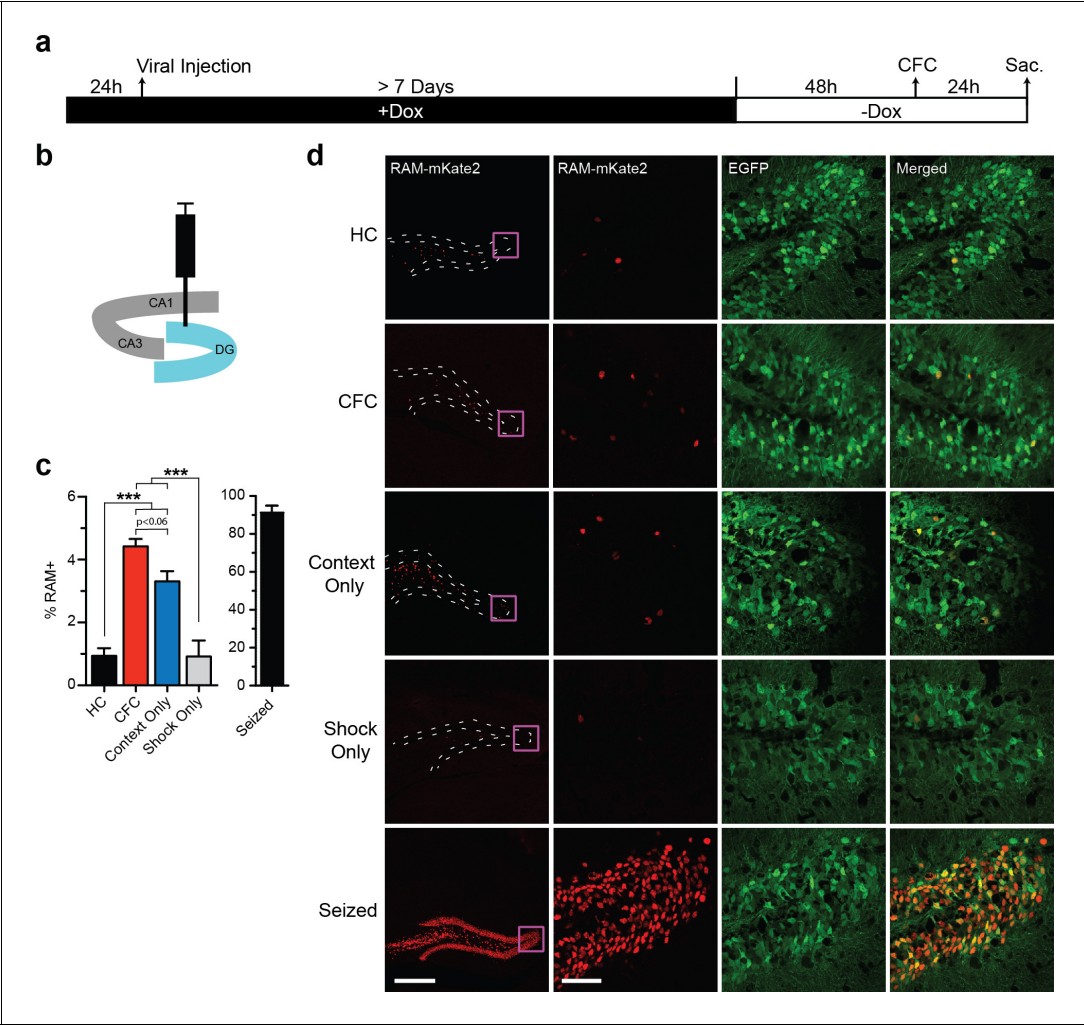

**Figure 3.** The RAM system labels active neuronal ensembles in the DG of the hippocampus. (a) Schematic timeline of the experimental procedure. Animals were kept on Dox diet (+Dox) from 24 hr before injection of AAV-RAM-NLS-mKate2 and AAV-Ef1α-EGFP into DG to a minimum of 7 days after the injection. Dox was then withdrawn (−Dox) 48 hr before either CFC, exposure to the novel context without shock (Context Only), receiving immediate shocks (Shock Only) or kainic acid treatment to induce seizures. Animals were sacrificed 24 hr later. A control cohort of similarly injected and treated animals was left undisturbed in their home cages (HC) for the entire period before sacrifice. (b) Schematic drawing of the hippocampus with the viral injection site (DG) highlighted in blue. (c) Percentage of RAM+ cells among total EGFP+ cells in the DG stratum granulosum for HC, CFC, Context Only, Shock Only and seized animals. The data for seized animals are replotted from *Figure 2c* (48 hr group). n = 6–9 animals per condition (with the exception for the Shock Only group, which consisted of 3 animals), one-way ANOVA, Tukey's post-hoc test. (d) Representative images of the DG region showing mKate2 (red) and EGFP (green) labeling for each of the HC, CFC, Context Only, Shock Only and seizure conditions. The scale bar is 300 μm for the left images and 50 μm for the three right columns of images. These images are enlarged from the areas marked by purple squares. All data in c are mean ± SEM. ***p<0.001.

The following figure supplements are available for figure 3:

**Figure supplement 1.** RAM labeling in the hippocampus following sensory experience is prevented by Dox diet.

**Figure supplement 2.** RAM labeling of CA3 pyramidal cells following contextual fear conditioning (CFC).

**Figure supplement 3.** RAM labeling following CFC persists for at least a week.

**Figure supplement 4.** Infection with RAM-channelrhodopsin (ChR2) or -archaerhodopsin (ArchT) prior to CFC enables light-activation or silencing, respectively, of repeated action potentials in RAM+ hippocampal DG granule cells.

*Kawashima et al., 2014*; *Xiu et al., 2014*), we chose to focus on active ensembles in the DG following CFC.

We co-injected AAV-RAM-NLS-mKate2 and AAV-Ef1α-EGFP into the DG and kept animals on Dox chow until 48 hr before CFC. The animals were then subjected to CFC in an initial context, context A, and the neuronal ensemble activated by this first exposure was labeled by RAM. Twenty-four hours later, animals were re-exposed to context A or to a novel context B. Animals were sacrificed 1.5 hr later and brain sections were stained for the IEGs FOS and NPAS4 to label ensembles triggered by the second context exposure (*Figure 4a–b*). Mice re-exposed to the conditioning context A froze significantly more than those exposed to the novel context B, suggesting that mice could distinguish between the two contexts (*Figure 4g*).

We quantified the degree of overlap between the RAM-labeled and IEG-labeled ensembles. If RAM specifically labels the neuronal ensemble encoding a particular context, we would expect to see more overlap after repeat exposure to the same context than after subsequent exposure to a new context. Indeed, the percentage overlap between RAM+ and either FOS+ or NPAS4+ granule cells (GCs) was significantly higher in animals re-exposed to the same context A than in those subsequently exposed to the new context B (*Figure 4c–f,h–i*). This suggests that RAM accurately labels neurons that are involved in processing specific context information. Additionally, the absolute percentage of RAM+ or IEG+ GCs was in the 2–4% range (*Figure 4h–i*), similar to previous results for this brain region following context exposure (*Garner et al., 2012*; *Ramirez et al., 2013*). These results indicate that the RAM system labels an active neuronal population similar to that identified by traditional IEG staining. Furthermore, the overlap percentage between RAM+ and IEG+ GCs for context A-A exposed animals was either on par with or significantly higher than overlap percentages shown in equivalent experiments from previously published studies. Furthermore, the roughly two-fold increase in overlap percentage between context A−A and context A−B exposed animals is also on par with previously published studies (*Denny et al., 2014*; *Ramirez et al., 2013*; *Deng et al., 2013*). This suggests that the RAM system can also be used for functional perturbation studies dependent on context discrimination (*Liu et al., 2012*; *Garner et al., 2012*; *Ramirez et al., 2013*; *Koya et al., 2012*; *Bossert et al., 2011*).

## Application of the RAM systems to another brain region - amygdala

We next tested the versatility of our RAM system by applying it to other brain regions. We injected AAV-RAM-NLS-mKate2 and AAV-Ef1α-EGFP into the lateral section (LA) of the basolateral amygdala (*Figure 5a–b*), which is the main input locus of the amygdala for auditory afferents. We examined RAM labeling in this region in animals subjected to tone-fear conditioning (TFC), a form of cued associative learning known to require the amygdala (*Pare and Duvarci, 2012*). Each cohort of animals was divided into the following four treatment groups: (1) left unperturbed in the home cage (HC), (2) TFC, (3) exposed to the tone alone without shock (Tone) and (4) received the immediate shock alone and removed from the testing chamber right away (Shock). Compared to HC controls, TFC led to a dramatic increase in the number of RAM+ neurons in the LA (*Figure 5c,f*). Interestingly, neither Tone nor Shock treatment alone resulted in RAM labeling significantly above the HC level (*Figure 5c,f*). The relative levels of RAM labeling in LA under each condition were very similar to those of IEG staining reported in published studies (*Senn et al., 2014*; *Liu et al., 2003*; *Radulovic et al., 1998*). Furthermore, long-term synaptic plasticity is known to occur in the LA following TFC to allow the formation of the tone-associated fear memory, while tone or shock alone does not result in long-term behavioral or synaptic changes (*Rogan et al., 1997*; *Rogan and LeDoux, 1995*). Our result (*Figure 5c,f*) therefore suggests that the neuronal ensembles labeled by RAM are likely involved in fear memory formation.

In the same set of the TFC experiment, we also targeted the basal nucleus (BA) of the BLA (*Figures 5a,6d–f*), which is a main target of LA and relays information from LA to the central amygdala (CeA). Like in the LA, TFC greatly increased the number of RAM labeled neurons in the BA, compared to HC (*Figure 5e–f*). In addition, the numbers of RAM labeled neurons in both Tone and Shock groups, although significantly lower than in the TFC group, were much greater than in HC (*Figure 5e–f*). Again, the relative levels of RAM labeling in BA were in agreement with those of IEG expression previously reported under similar conditions (*Senn et al., 2014*; *Liu et al., 2003*; *Radulovic et al., 1998*).

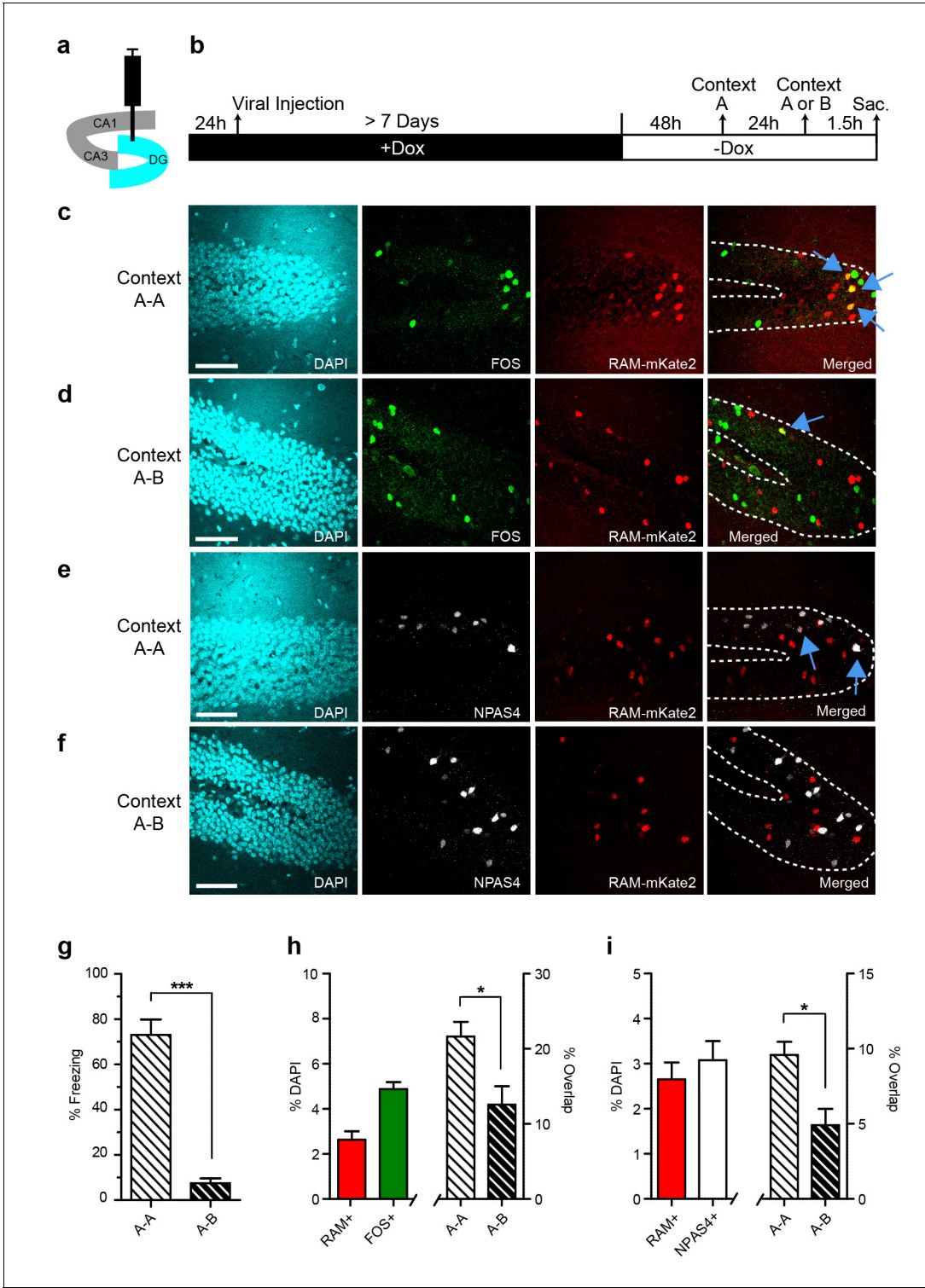

**Figure 4.** Contextual memory recall preferentially reactivates cells initially labeled with RAM during memory encoding. (**a**) Schematic drawing of the hippocampus with the viral injection site (DG) highlighted in blue. (**b**) Timeline of the experimental procedure. Animals were injected with AAV-RAM-NLS-mKate2 and kept on a Dox diet (+Dox) for at least 7 days after surgery. Two days after Dox removal (−Dox), animals were exposed to context A (for RAM labeling) and shocked, and then exposed to either context A again or a new context B 24 hr later (for IEG labeling). Animals were sacrificed 1.5 hr after the second context exposure. (**c, d**) Representative images of DG after A−A (**c**) and A−B (**d**) exposure showing DAPI staining (cyan), FOS staining (green), RAM labeling (red), and the merged image. (**e, f**) As panels **c** and **d**, except staining with NPAS4 (white) instead of FOS. (**g**) Freezing behavior observed during re-exposure to context A or B. n = 4–5 animals per condition, Student's t-test. (**h**) Percentage of all DAPI-labeled cells (i.e. all

*Figure 4 continued on next page*

*Figure 4 continued*

neurons) in the DG stratum granulosum labeled with RAM (red) and FOS (green; n = 9 animals per condition), and the percentage overlap between the RAM and FOS labeled cells following A-A exposure (n = 4 animals) and A-B exposure (n = 5 animals). Student's t-test. (**i**) As **h**, except with NPAS4 (white) instead of FOS. The scale bar is 50 µm for all images. Data in **g–i** are mean ± SEM. *p<0.05, ***p<0.001.

Because our TFC condition setup also contained the novel context component, as animals were not habituated to the testing chamber, we also examined RAM labeling in the CeA, a region that has been shown to have increased IEG expression following fear conditioning (*Radulovic et al., 1998*; *Hall et al., 2001*; *Milanovic et al., 1998*; *Day et al., 2008*). In animals subjected to TFC, the neuronal ensembles labeled by RAM in the CeA were significantly larger compared to HC (*Figure 5g–i*). Taken together, these results indicated RAM can label neuronal ensembles that are important for fear memory formation in the amygdala.

## Application of the RAM system to other species

One major advantage of the RAM system is its applicability to model organisms other than the mouse. We therefore tested our AAV-RAM system and $P_{RAM}$ in rats and flies, respectively. AAV-RAM-NLS-mKate2 and AAV-Ef1α-EGFP was injected into the infragranular layers (V and VI) of the rat medial prefrontal cortex (mPFC) and animals were placed on Dox (*Figure 6a–b*). Five days after injection, rats were taken off Dox for four days before being subjected to the inescapable stress (IS) paradigm, which is known to engage the mPFC (*Wang et al., 2014*). A significantly higher percentage of RAM+ labeling was observed in animals subjected to IS compared to HC controls (*Figure 6c–d*). These results indicate that the RAM system can be used successfully in species other than mouse, and also in an additional brain region (the mPFC).

Since *Drosophila* contains both *Fos−* and *Npas4*-like transcription factors (*Perkins et al., 1988*; *Jiang and Crews, 2007*), we tested $P_{RAM}$ in flies using a strategy that has been successfully implemented for CRE-based reporters (*Zhang et al., 2015*; *Tanenhaus et al., 2012*). We generated a transgenic $P_{RAM}$-luciferase (RAM-luc) reporter fly in which luciferase expression is tightly controlled by both the $P_{RAM}$ promoter and FLP-recombinase (*Figure 6e–f*, and see 'Methods'). To test RAM-luc expression and its dependence on neuronal activity, the RAM-luc reporter transgene was combined with a transgene expressing Flp recombinase in all adult neurons (using the pan-neuronal 232B GAL4 driver). Luciferase activity measured in live flies over time showed high expression in the adult *Drosophila* brain. Most importantly, RAM-luc reporter activity was circadian (*Figure 6g*), suggesting that $P_{RAM}$ functions as a robust activity-dependent promoter in *Drosophila*.

To assess whether learning experience can alter RAM-luc activity, pan-neuronal RAM-luc flies were trained in an olfactory memory task (Forward Spaced, or FS, training, in which an odor was paired with shock presentation) that is known to produce robust long-term memory that lasts up to a week (*Zhang et al., 2015*). Two days after training, RAM-luc activity in FS-trained flies was significantly higher than in those in the control group exposed to un-paired odor and shock presentation (Backward Spaced, or BS, training; *Figure 6h*). Although further work is needed to completely validate the RAM-luc reporter fly in behavioral studies, these results suggest that it can be used to identify memory-specific cellular populations in *Drosophila*.

## Application of the RAM system to other cell types and Cre-dependent RAM

We next examined the ability of our RAM system to specifically label cell types other than principal neurons that are transcriptionally induced by experience. We first sought to confirm that the RAM system works well in GABAergic neurons, which was implied by RAM labeling of neurons in CeA (*Figure 5g–i*), a region comprised of 95% GABAergic neurons. We injected rats with AAV-RAM-NLS-mKate2 and AAV-Ef1α-EGFP into mPFC and subjected the animals to the IS paradigm (*Figure 7a–b*). Indeed, we found that the number of GAD67-labeled GABAergic neurons that were RAM+ was significantly higher in animals subjected to IS than the HC controls (*Figure 7c–d*). A small fraction of these induced RAM+ GABAergic neurons were parvalbumin-expressing (PV+) interneurons (*Figure 7—figure supplement 1a–d*). In mice, CFC treatment resulted in RAM labeling in a substantial number of somatostatin-positive GABAergic neurons in the hilus of the DG (*Figure 7—figure*

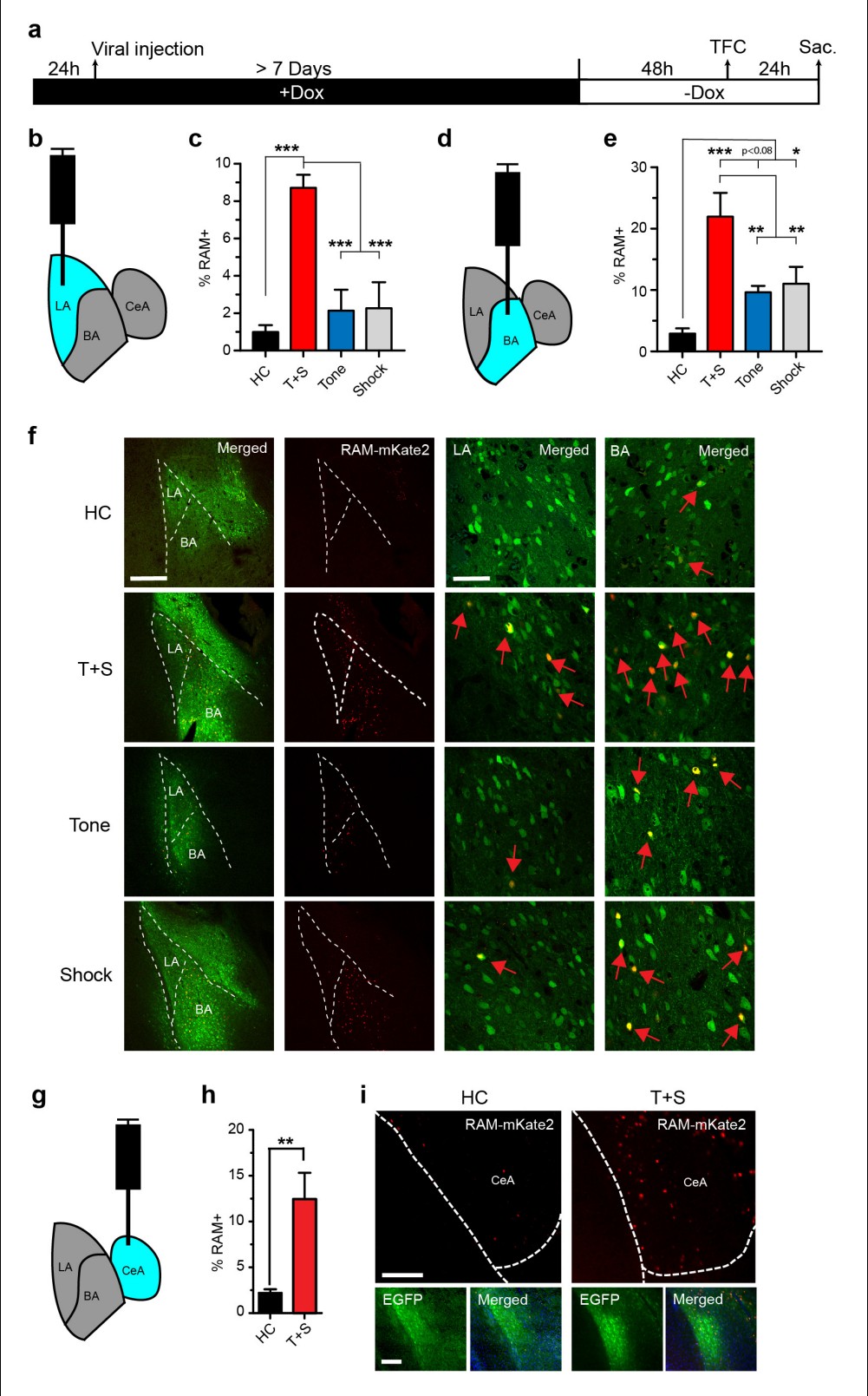

**Figure 5.** The RAM system labels active neuronal ensembles in the amygdala. (a) Schematic timeline of the experimental procedure. While on a Dox diet (+Dox), AAV-RAM-NLS-mKate2 and AAV-Ef1α-EGFP vectors were injected into the lateral amygdala (LA, b), basal nucleus (BA, d), or central amygdala (CeA, g). After at least 7 days, Dox was removed (-Dox) for 48 hr, the animals were exposed to tone-fear conditioning (TFC; consisting of Tone

*Figure 5 continued on next page*

*Figure 5 continued*

and Shock, T+S), Tone only, or Shock only and sacrificed 24 hr later. A control cohort of similarly injected and treated animals was left undisturbed in their home cages (HC) for the entire period before being sacrificed. (**b**) Schematic drawing of the amygdala with the injection and the quantification site (LA) highlighted in blue. (**c**) Percentage of RAM+ cells among total EGFP+ cells in LA for HC, T+S, Tone, and Shock animals. n = 3–6 animals per condition, one-way ANOVA, Tukey's post-hoc test. (**d**) As panel **b**, but for the BA region. (**e**) Percentage of RAM+ cells among total EGFP+ cells in BA for HC, T+S, Tone and Shock animals. n = 3–4 animals per condition, one-way ANOVA, Tukey's post-hoc test. (**f**) Representative images of neurons labeled in LA and BA for HC, T+S, Tone and Shock conditions. Neurons labeled by AAV-RAM-mKate2 are red and cells labeled by AAV-Ef1α-EGFP are green. The merged left image is enlarged in the two right images. Red arrows indicate RAM and EGFP double-labeled cells. The scale bar is 300 μm and 50 μm for left and right images, respectively. (**g**) As panel **b**, but for the CeA region. (**h**) Percentage of RAM+ cells among total EGFP+ cells in CeA for HC and T+S animals. n = 3 animals per condition, Student's t-test. (**i**) Representative images of neurons labeled in CeA region. The scale bar is 150 μm for all images. All data in **c**, **e** and **h** are mean ± SEM. *p<0.05, **p<0.01, ***p<0.001.

supplement 2a–f). Additionally, using the Cre-dependent FLEX system and luciferase assay, we confirmed that $P_{RAM}$ was induced by KCl stimulation in cultured cortical neurons derived from Gad2-Cre mice wherein Cre is expressed exclusively in neurons expressing the GABAergic neuron marker Gad65 (*Figure 7e–g*). These results confirm that the RAM system achieves robust activity-dependent labeling of GABAergic neurons.

With a view to future investigation of ensembles involving specific cell types, we developed a Cre-dependent RAM (CRAM) system, in which the effector gene can only be expressed in cells that express Cre (*Figure 7h*). When CRAM-tdT virus was injected into the DG of Gad2-Cre mice, only GABAergic neurons (most prominently somatostatin-expressing neurons in the hilus) and none of the glutamatergic granule cells were labeled with tdT after kainic acid-induced seizure (*Figure 7—figure supplement 3a–d*). While IEGs are known to be induced by experience in GABAergic neurons (*Spiegel et al., 2014*), active ensembles of GABAergic neurons have not been explored. Using Cre driver mouse lines for various types of neurons, we envision the CRAM system as a tool well suited to investigating active ensembles of GABAergic neurons and other specific neuronal cell types.

Like the RAM system, CRAM also worked well in the rat, in this case to identify neuronal ensembles with specific anatomical connections. Since a major projection of the rat PFC is the dorsal striatum (*Gabbott et al., 2005*), we injected a retrograde canine adenovirus that expresses Cre recombinase into the dorsomedial striatum (DMS) to deliver Cre to DMS-projecting mPFC neurons. AAV-CRAM-tdT was subsequently injected into the mPFC and the animals were subjected to IS treatment (*Figure 7i–j*). We observed significantly more tdT-labeled neurons in animals subjected to IS compared to HC controls (*Figure 7k–l*), suggesting CRAM successfully labeled IS-activated mPFC neuronal ensembles that project to the DMS.

## Discussion

Here we present the RAM (Robust Activity Marking) system, which makes possible the sensitive and specific labeling and manipulation of the active ensembles of neurons associated with a designated sensory and behavioral experience. Compared to existing methods used to gain access to recently active neuronal ensembles (*Guenthner et al., 2013*; *Reijmers et al., 2007*; *Smeyne et al., 1992*; *Eguchi and Yamaguchi, 2009*; *Kawashima et al., 2013*), the RAM system delivers unprecedented versatility and much improved selectivity. Additionally, the system showcases a novel, informatics-based approach to constructing small synthetic IEG-sensitive promoters. We believe that this approach could be used to generate other small yet highly sensitive promoters that, when combined with the RAM platform, would allow in vivo interrogation of specific activity-dependent genetic programs.

The most important features of a neuronal-activity reporter system are high sensitivity and selectivity, resulting from a combination of low background labeling and robust effector gene expression in the active ensemble associated with a designated experience. RAM achieves exceptionally high selectivity and sensitivity through the use of an optimized activity-dependent promoter with the highest fold induction we have observed among existing small activity-regulated promoters, paired

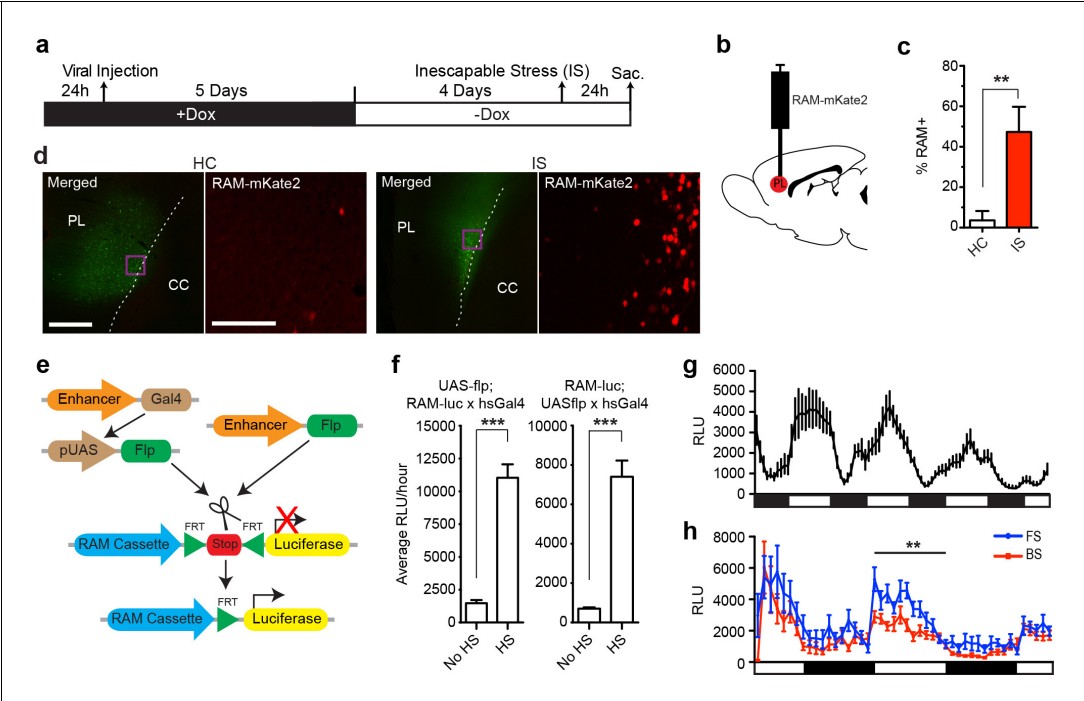

**Figure 6.** The RAM system in rats and flies. (**a–d**) RAM labels active neuronal ensembles in rats exposed to inescapable stress (IS). (**a**) Schematic timeline of the experimental procedure. Rats were injected with AAV-RAM-NLS-mKate2 and AAV-Ef1α-EGFP vectors in the medial prefrontal cortex (mPFC). After 10 days they were either subjected to IS or left undisturbed in their home cages (HC). (**b**) Schematic drawing of the rat brain with the red region indicating the target area (PL: prelimbic cortex of the mPFC) for virus infection and quantification of RAM+ cells. (**c**) Percentage of RAM+ cells among total EGFP+ cells in the prelimbic cortex in HC and IS animals. Data are mean ± SEM, n = 3–4 animals per group, Student's t-test, **p<0.01. (**d**) Representative images of prefrontal cortex showing mKate2 (red) and EGFP (green) fluorescence in rats subjected to IS or HC conditions. Areas in purple squares are enlarged in the right image for each condition. CC: corpus callosum. Scale bars are 500 and 100 μm for the left and right images, respectively. (**e–h**) The *Drosophila* RAM reporter system. (**e**) Schematic diagram of the *Drosophila* RAM reporter system. The RAM-luc transgene can be turned on in specific cell types by the targeted expression of Flp recombinase using the GAL4-UAS system (left) or a cell-type specific driver (right). (**f**) *Drosophila* RAM reporter activity has low baseline levels and high fold induction. The specificity of the RAM-luc to Flp recombinase was tested using a UAS-Gal4 system, in which Flp recombinase expression is under the control of a heat-shock HS promoter. Flies in the no heat-shock (No HS) condition were maintained at 20°C throughout development and experimental conditions. For the heat-shock (HS) condition, flies were exposed to a 37°C heat shock for 30 min and allowed to recover for a full day at 20°C before measuring reporter expression. To ensure that results were not due to insertional effects, the UAS-flp transgene was combined with fly lines with the reporter transgene on either chromosome II (RAM-luc;UAS-flp) or chromosome III (UAS-flp;RAM-luc). n = 40–47 flies per group, Student's t-test, ***p<0.001. (**g**) Pan-neuronal RAM-luc reporter expression displays circadian rhythm. The RAM-luc reporter transgene was combined with a transgene expressing FLP recombinase in all adult neurons and luciferase activity measured in live flies over time. Bars under plots indicate day (light) and night (dark). (**h**) Pan-neuronal RAM-luc reporter expression is sensitive to memory formation in *Drosophila*. Flies as described in **g** were trained in an olfactory memory task. 24 hr after training, flies exposed to Forward Spaced (FS) training showed significantly higher RAM-luc expression than control flies exposed to Backward Spaced (BS) training. Bars under plots indicate day (light) and night (dark). n = 23–24 flies per group, Student's t-test, **p<0.01.

with an improved version of the Tet-Off system. In vivo RAM achieved 37-fold and five-fold increases in labeling after CFC, compared to HC controls, in CA3 and DG, respectively. This is significantly greater than the two–three-fold induction achieved using existing technologies (*Guenthner et al., 2013*; *Ramirez et al., 2013*). The five–37-fold induction achieved with the RAM system implies that 80–97% of the labeled neurons are truly associated with the specific experience, compared to 50–67% with existing technologies, representing a dramatic improvement in signal-to-noise ratio. This improvement alone means that the RAM system can be used in many brain regions and behavioral applications where existing methods will not work.

With its high sensitivity and selectivity, the RAM system will be ideal for functional perturbation experiments in which the activity of ensemble neurons, identified by RAM, can be manipulated through the targeted expression of effector molecules such as opsins. We show the expression of opsins in CFC-induced, RAM-labeled ensemble neurons can be activated or inhibited 24 hr after

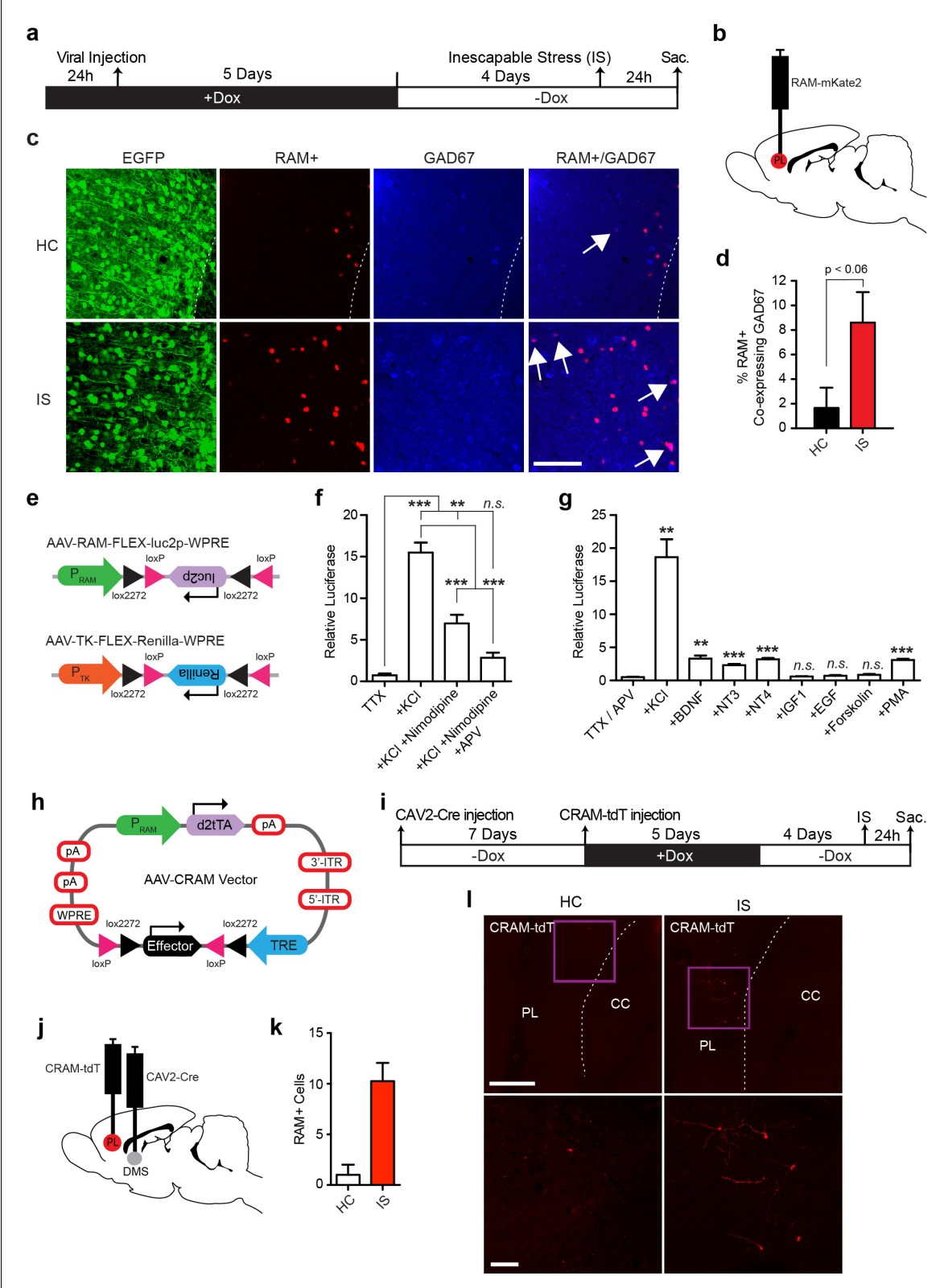

**Figure 7.** RAM labeling of transcriptionally active interneurons and application of Cre-dependent RAM (CRAM). (a–d) RAM labels active GAD67+ neurons in rats exposed to inescapable stress (IS). (a) Schematic timeline of the experimental procedure. Rats were injected with AAV-RAM-NLS-mKate2 and AAV-Ef1α-EGFP in the mPFC. After 10 days they were either subjected to IS or left undisturbed in their home cage (HC). (b) Schematic drawing of the rat brain with the red region indicating the target area (PL: prelimbic cortex of the mPFC) for viral infection and quantification of RAM+ cells. (c) *Figure 7 continued on next page*

*Figure 7 continued*

Representative images of prefrontal cortex showing EGFP (green), mKate2 (red) and GAD67 (blue) fluorescence in rats subjected to IS or HC conditions. White arrows indicate RAM+ and GAD67+ double-labeled cells. The scale bar is 100 µm and applies to all images. (**d**) Percentage of RAM+ cells co-labeled with GAD67+ in the prelimbic cortex of HC and IS animals. Data are mean ± SEM, n = 3–4 animals per group, Student's t-test. (**e–g**) Neuronal activity-dependence of the $P_{RAM}$ promoter in GABAergic neurons. (**e**) Design of AAV vectors used for luciferase assays in dissociated neuronal cultures from Gad2-Cre transgenic mice. The reading frames of luciferase (luc2p) and renilla are double inverted and flanked by double loxP sites (FLEX) and inserted downstream of the $P_{RAM}$ and thymidine kinase promoter ($P_{TK}$) respectively. (**f**) Relative luciferase activity of $P_{RAM}$ in Gad2-Cre cells following KCl stimulation, with application of Nimodipine and/or APV. n = 3 separate experiments per condition, one-way ANOVA, Tukey's post-hoc test. (**g**) Relative luciferase activity of $P_{RAM}$ in Gad2-Cre cells after application of various neurotrophic factors and drugs. n = 3 separate experiments per condition, Student's t-test. (**h**) Schematic diagram of the AAV-CRAM construct. The effector gene is flanked by double loxP sites (FLEX). (**i–l**) CRAM labels IS-activated mPFC neurons projecting to the dorsomedial striatum (DMS). (**i**) Experimental procedure. Rats were injected with CAV2-Cre in the DMS and AAV-CRAM-tdT in the PL (**j**), then either subjected to IS or left undisturbed in their home cages (HC). (**k**) Number of CRAM+ cells in the prelimbic cortex of HC and IS animals. Data are mean ± SEM, n = 2 animals per group. (**l**) Representative images of the prefrontal cortex showing tdT fluorescence in rats subjected to IS or HC. Areas in purple squares are enlarged in the lower images. CC: corpus callosum. Scale bars are 500 and 100 µm for the upper and lower images, respectively. Data in **d**, **f**, **g** and **k** are mean ± SEM. **p<0.01, ***p<0.001.

The following figure supplements are available for figure 7:

**Figure supplement 1.** RAM labeling of transcriptionally active parvalbumin (PV) cells after inescapable stress (IS).

**Figure supplement 2.** RAM labeling in the DG hilus.

**Figure supplement 3.** Validation of the AAV-CRAM system.

**Figure supplement 4.** FOS expression in parvalbumin (PV) and somatostatin (SST) positive cells in primary visual cortex (V1) and dentate gyrus (DG) of the hippocampus.

CFC (*Figure 3—figure supplement 4a–j*), demonstrating the feasibility of such an approach using RAM. Based on the robust results we obtained by applying RAM to label ensembles in several brain regions (DG, CA3, amygdala and mPFC) using appropriate behavioral paradigms (CFC, TFC and IS), we anticipate that RAM can be readily used to investigate neuronal ensembles in other brain regions, such as CA1, striatum, nucleus accumbens, hypothalamus, etc., for a variety of behavioral manipulations in the future.

Like all other existing activity reporters currently being used in the neuroscience field, the sensitivity of the RAM system has not been systematically measured. For example, we do not know the activation threshold for RAM. More specifically, what is the minimal amount of neuronal activity required to activate RAM? What type of neuronal activity is preferentially captured by RAM? How well can RAM distinguish information-carrying neuronal firing from fluctuating noise? Future experiments aimed to obtain these basic operational parameters for RAM, and for any activity reporter, will be critical for us to be able to use these reporters to gain mechanistic understanding of how sensory and behavioral information is processed in the brain.

As for any IEG-based activity reporter, the efficiency of RAM labeling in particular brain regions or of particular cell types depends on their propensity to express IEGs. For example, activated PV+ neurons appear to be less likely to express IEGs. Even after pentylenetetrazol (PTZ)-induced seizure, only about 20% of the PV+ neurons, compared to over 80% of somatostatin expressing (SST+) neurons, in the cortex were transcriptionally activated to express FOS (*Figure 7—figure supplement 4a–b*). A similar trend was observed in DG after KA-induced seizure (*Figure 7—figure supplement 4a, c*), suggesting either PV+ neurons may be less likely to be recruited to active ensembles or they may require non-IEG-based approaches to be effectively labeled.

The compact design of the RAM system allows it to be packaged into a single AAV while still accommodating an effector gene up to 1.8 Kb in size. The RAM system can readily accommodate most of the effector genes commonly used to label or manipulate neural circuits. We envision future experiments using effector genes for genetic profiling (e.g. RIBO-TRAP [*Sanz et al., 2009*]), rabies-mediated monosynaptic anatomical tracing (*Callaway, 2008*), and functional perturbation of behavior (*Liu et al., 2012*) (see *Figure 3—figure supplement 4e,j*). In cases where other, larger promoters

and/or larger effector genes are needed, the RAM system can split between two AAVs, one containing the d2tTA expression cassette and the other the TRE-effector cassette.

Cre-dependent RAM (CRAM) extends the versatility of the RAM system even further. For example, CRAM can be used to study active neuronal ensembles of a particular cell type when Cre expression is controlled by a cell type-specific promoter in a mouse line or virus. Also, by delivering Cre via retrograde and anterograde viruses, neuronal ensembles with specific anatomical connectivities can be explored. Finally, a Flp-dependent RAM system could easily be developed, allowing additional applications of the system.

Since the $P_{RAM}$ 10mer is highly enriched in activity-dependent enhancers (*Supplementary file 1*) and enhancers are mostly conserved between species (*Kim et al., 2010*), $P_{RAM}$ is likely to be active in many different species. Here we demonstrate that $P_{RAM}$ works in mice, rats and flies, but it may be applicable to other species as well. Given the flexible design of the RAM system and the relative ease of AAV production, the system makes possible many new and powerful experiments in a field with relatively few viable tools (*Cruz et al., 2013*; *Kawashima et al., 2014*).

## Methods

### Screening for enriched enhancer

The coordinates on the mouse chromosome mm9 assembly for the 11,830 activity regulated enhancers identified by *Kim et al. (2010)* were downloaded and the sequences from a 2 Kb window centered at each enhancer were extracted. We first looked for DNA elements enriched among activity-regulated enhancers. Sequences in a 160 bp window centered on each enhancer were extracted and then used as inputs for the *de novo* motif finding software Weeder (*Pavesi et al., 2004*) (http://159.149.160.51/modtools/). The top motif identified strongly resembles the consensus for AP-1 (*Figure 1—figure supplement 1*).

To calculate the enrichment of all AP-1-containing 10mers (128 in total, see *Supplementary file 1*), we searched for perfect matches on both strands for each 10mer and counted the number of hits on each 40 bp non-overlapping bin. Enrichment for each 10mer was calculated as the total number of occurrences in the four central bins (i.e. the −80 to +80 region) divided by the total number of occurrences in the eight furthest bins on each flank (−1000 to −681 and +681 to +1000). The larger flanking regions were used to obtain a more accurate estimate of the fold enrichment at the enhancer center relative to the flanks. The enrichment factor, E, adjusted for the larger size of the flanking bins, was defined as $E = 4*C/F$, where C is the number of hits in the central bins and F is the number of hits in the flanks. The significance of finding each motif was calculated using a binomial test (in MATLAB) under the null hypothesis that the hits should be equally distributed between the center and flanks. The p-value is given by $p=1-CDF (4*C, 4*C + F, 0.5)$, where CDF (n, N, 0.5) is the cumulative distribution function of finding n elements from a binomial distribution with parameters N and 0.5. If there were no hits in either the center bins or the flanks, the p-value was set to 0.5, and the motif was not considered further. The motifs were ranked by their enrichment factors. Motifs that did not have a Bonferroni corrected p-value<0.05 were excluded from further analysis. A full list of 10mers containing the AP-1 consensus site, ranked by their enrichment factors, can be found in *Supplementary file 1*. Several enriched 10mers (E1-E3) were selected to construct synthetic promoters for further testing based on their enrichment factor and the associated p-value.

### Cloning of plasmids

All luciferase-encoding plasmids were based on the pGL4.11 backbone (luc2p, Promega, Madison, WI). Complementary strands of enhancer modules (EM), $P_{NRE+AP1}$, $P_{RAM}$, enriched 10mer-containing enhancer module variants (E1-E3) and CME sequences, each containing four identical repeats, were ordered as DNA oligos (Sigma Aldrich, Natick, MA) with the following sequences and annealed:

$P_{NRE+AP-1}$: 5′ − CT**TCGTG**ACTAGTCT**TGACTCA**GA −3′
$P_{RAM}$:    5′ − CTAGAAGTTTGT**TCGTGACTCA**GA −3′
E1:    5′ − CTAGAAGTTTGT**TGACTCACCC**GA −3′
E2:    5′ − CTAGAAGTTTGT**TGACTCATTA**GA −3′

```
E3:    5' – CTAGAAGTTTGTGTATGACTCAGA –3'
CME:   5' – CTAGAAATTTGTACGTGCCACAGA –3'
```

Nucleotides underlined (in bold) denote putative binding motif and/or variable flanking regions. One annealed $P_{RAM}$ EM served as a template for making various enhancer repeats (1, 2, 3, 4 and 8 enhancer repeats) by In-Fusion cloning (Clontech). The minimal promoters were PCR cloned from other constructs: the *FOS* minimal promoter (99 bp) from pOF-luc plasmid (*Selvaraj and Prywes, 2003*); the *Arc* minimal promoter (421 bp) from ESARE plasmid, a gift kindly provided by Haruhiko Bito (*Kawashima et al., 2013*); the human beta-globin (hBG) minimal promoter (49 bp, a gift kindly provided by Guoping Feng), the cytomegalovirus (CMV) minimal promoter (226 bp) from pcDNA3 plasmid (Invitrogen). Annealed enhancers were inserted upstream of the minimal promoter(s) and then inserted into pGL4.11 using Kpn1 and Nhe1 sites.

Other activity-dependent promoters, inserted into pGL4.11 to drive luciferase expression, were derived from other studies: *Npas4* (*Ramamoorthi et al., 2011*), p1BDNF (*Ramamoorthi et al., 2011*), ESARE (kindly provided by Haruhiko Bito [*Kawashima et al., 2013*]), *Fos* (*Barth et al., 2004*), CREB reporter from CRE-luc2p (Promega), and MEF2 reporter (*Selvaraj and Prywes, 2003*). Over-expression plasmids encoding *Npas4* and *Fos* were constructed by PCR cloning as follows: the coding regions of *Npas4* and *Fos* were inserted into the pEF1/*myc*-His (A) backbone (Life Technologies, Carlsbad, CA) using Spe1 and Pme1 sites. The empty pEF1/*myc*-His (A) backbone served as a transfection control.

Vectors used to engineer and characterize the RAM driven Tet-OFF system were derived from pTet-Off-Advanced and pTRE-Tight vectors (Clontech, Mountain View, CA). The CMV promoter in pTet-Off-Advanced was replaced by inserting $P_{RAM}$ within the Spe1 and Sac1 sites ($P_{RAM}$-tTA). Decreased stability tTA was made by fusing the degradation domain of mouse ornithine decarboxylase (MODC, derived from pD2EGFP; Clontech) to the N-terminus of tTA, making $P_{RAM}$-d2tTA. The MODC contains a PEST sequence that enhances tTA degradation (*Li et al., 1998*). Transferring luciferase (luc2p) from pGL4.11 into pTRE-Tight using HindIII and Xba1 sites resulted in the TRE luciferase (pTRE-luc2p) vector.

## Glia cultures and neuron-glia co-cultures

Preparation of cultures with a confluent glial monolayer has been described elsewhere (*Paradis et al., 2007*). Briefly, astrocytes derived from P1-P2 rat cortices were plated at low density in DMEM + 10% FBS on 12 mm glass coverslips coated with poly-D-lysine and laminin in 24-well plates, and stored and maintained at 37°C in a humidified incubator with 10% $CO_2$. Once the plated glia cells had formed a confluent monolayer, typically after 7 days, dissociated hippocampal neurons from mouse P1 pups (C57Bl/6, Charles River Laboratory) were plated at a density of 40,000 cells per well, and the media was changed to NBA supplemented with B27 (Invitrogen, Carlsbad, CA) and GlutaMAX (Life Technologies). AraC (5 µM, Sigma) was added the next day and conditioned media was supplemented with fresh media every fifth day. Pure glia cultures were used for a set of luciferase experiments, whereas neuron-glia co-cultures were used for immunocytochemistry (ICC) experiments. For ICC experiments, cultures were stimulated with bicuculline (50 µM, Sigma Aldrich) and 4AP (250 µM, Tocris, UK), and doxycycline was applied at 40 ng/mL final concentration.

## Immunocytochemistry

Cultures were washed once with PBS (pH 7.4), then fixed for 10 min with 4% paraformaldehyde in PBS containing 0.1% Triton X-100. Subsequently, cultures were washed 3 times with 10 mM glycine in PBS followed by a final rinse with PBS. The primary antibody was added to a solution of 0.1% porcine gelatin (Sigma), 0.25% Triton X-100, 0.23 M NaCl, and 0.015 M phosphate buffer pH 7.4, and coverslips were then transferred to a humidifying chamber. Antibody solution was dropped (50 µL) onto each coverslip and allowed to incubate at 4°C overnight. The next day, the coverslips were washed 4 times with PBS, and incubated in 50 µL of secondary antibody solution for 1 hr at room temperature. Coverslips were washed twice with PBS and mounted on Superfrost slides using DAPI Flouromount-G (SouthernBiotech, Birmingham, AL) before imaging and analysis. The primary antibodies were MAP2 (mouse, 1:1000, Sigma, M9942) and GFAP (mouse, 1:1000 Sigma, G3893). The secondary antibody was Alexa Fluor 488 (Goat anti-mouse IgG, 1:1000, Life Technologies).

## Cell cultures for luciferase assays

Mouse pups, C57Bl/6 (Charles River Laboratory) or Gad2-Cre (*Gad2*$^{tm2(cre)Zjh}$/J, Jackson Laboratory), at P1 were used to prepare dissociated hippocampal cultures as previously described (*Lin et al., 2008*). Cultures were plated at 100,000 cells per well on 24-well plates coated with poly-D-lysine. Cultures were incubated in Neurobasal A Medium (NBA, Life Technologies) with horse serum and glutamine added. The cultures were maintained at 37°C in a humidified incubator with 5% $CO_2$. After three hours of incubation, the medium was changed to NBA supplemented with B27 (Invitrogen) and GlutaMAX (Life Technologies). Neurons were transfected using lipofectamine 2000 (Life Technologies) on DIV5. Reporter plasmids (in pGL4.11) expressing firefly luciferase and a plasmid expressing renilla luciferase under control of the constitutive thymidine kinase promoter (TK)-Renilla (Promega) were co-transfected in every experiment. For a subset of luciferase experiments, cultures were co-infected with AAV vectors (0.1 µL per well) expressing firefly luciferase and TK-Renilla, respectively.

On the day of stimulation, TTX and APV were typically added 1 hr prior to stimulation, and cells were stimulated for 6 hr with 35 mM KCl (but see *Supplementary file 2* for precise details) before being rinsed briefly in PBS and lysed in passive lysis buffer (Promega). The Dual-Glo Luciferase Assay System reagents were used according to the manufacturer's instructions (Promega). Firefly luciferase levels were measured and expressed relative to renilla luciferase levels. Data were compiled from separate experiments each conducted in triplicate, and repeated a minimum of three times. Fold induction of luciferase was calculated as the ratio between stimulated and unstimulated conditions.

The primary criterion we used to evaluate the synthetic promoters for use as neuronal activity reporters was the fold change in luciferase expression. This is the ratio of the relative luciferase values between stimulated and unstimulated conditions and represents the activity-dependence and selectivity of the promoter. A secondary criterion was the absolute expression level after stimulation, which is the absolute luciferase value normalized against an internal control and reflects the absolute transcriptional strength of the promoter. It is possible for a weak promoter to have high fold induction, due to extremely low activity under basal unstimulated conditions, but to only drive transcription modestly, which will limit its ability to drive the expression of an effector gene to sufficient levels to achieve the required effects.

Drugs and neurotrophic factors, typically applied together with 10 µL NBA into the wells, were used at the following final concentrations: TTX (1 µM, Tocris), APV (100 µM, Tocris), Nimodipine (5 µM, Tocris), Bicuculline (50 µM, Sigma Aldrich), 4AP (250 µM, Tocris), recombinant human BDNF (50 ng/µL, PeproTech, Rocky Hill, NJ), recombinant human NT3 (50 ng/µL, PeproTech), recombinant human NT4 (50 ng/µL, PeproTech), recombinant human IGF1 (50 ng/µL, PeproTech), recombinant human EGF (50 ng/µL, PeproTech), Forskolin (10 µM, Tocris) and PMA (100 ng/µL, Tocris).

## Construction of RAM viral vector

The AAV vector V032 (pFB-AAV-CMV-WPRE-SV40pA, kindly provided by Rachael Neve) served as the backbone for constructing the AAV-RAM vector. All components within the 5'-ITR and 3'-ITR defined by the Mlu1 and Kpn1 sites, respectively, were first removed from V032. At the 5'-ITR site, the TRE promoter followed by a multiple cloning site (MCS), WPRE and bGH poly(A) signal was inserted as one expression cassette in the 5' to 3' direction. In the same orientation, the second expression cassette, consisting of $P_{RAM}$ driving d2tTA followed by a SV40 poly(A) signal, was inserted downstream of the first cassette and terminated by the 3'-ITR. The two expression cassettes were insulated by incorporation of synthetic poly(A) and RNA polymerase II transcriptional pause signals (both derived from pGL4.11) between the two cassettes. Fluorescent proteins, including tdTomato and mKate2, and the two opsins, Channelrhodopsin (ChR2, a kind gift from Gloria Choi) and Archaerhodopsin (ArchT, a kind gift from Ed Boyden), were inserted into the MCS. The plasmids AAV-RAM and AAV-CRAM vectors (as depicted in *Figure 1f*, pAAV-RAM-d$_2$TTA-pA::TRE-MCS-WPRE-pA; and *Figure 7h*, pAAV-RAM-d$_2$TTA-pA::TRE-FLEX-MCS-WPRE-pA) with an empty MCS at the effector gene position have been deposited at Addgene. The transfection control plasmid, pAAV-Ef1α-EGFP-WPRE-pA, was made by replacing mVenus in pAAV-Ef1α-mVenus-WPRE-pA (kindly provided by Jonathan Ting) using BamHI and BsrGI sites.

## Viruses and viral production

For ICC, we used AAV-RAM-tdTomato (AAV-RAM-d2tTA-pA::TRE-TdTomato-WPRE-pA, AAV8 sero-type, Virovek, Hayward, CA, 2.21E13 vg/mL). For rat studies, we used AAV-RAM-NLS-mKate2 (AAV-RAM-d2tTA-pA::TRE-NLS-mKate2-WPRE-pA, AAV1 serotype, Virovek, 2.18E13 vg/mL) and transfection control AAV-Ef1α-EGFP (AAV-Ef1α-EGFP-WPRE-pA, AAV1 serotype, UPenn Vector Core, 5.51E13 vg/mL). We also generated homemade AAVs using a FuGene6 (Promega) mediated triple plasmid co-transfection method in HEK293t cells. Three days after transfection, cells were harvested and virus was purified using an adapted Iodixanol gradient purification protocol (*Matsui et al., 2012*). These AAVs were a mixture of AAV2/2 (rep/cap) and AAV2/8 serotypes at a 1:1 ratio. Additionally, our AAV2/2 capsid incorporated three mutations (Y444F, Y500F and Y730F) (*Mowat et al., 2014*) while our AAV2/8 capsid had a double mutation (Y447F and Y733F) (*Qiao et al., 2012*). To determine the properties of AAV-RAM expression and for quantification of behaviorally labeled cells and IEG overlap experiments in vivo, we used homemade AAV-RAM-mKate2 (AAV-RAM-d2tTA-pA:: TRE-NLS-mKate2-WPRE-pA, mixed AAV2/8 serotype, 1.28E13 vg/mL) and transfection control AAV-Ef1α-EGFP (AAV-Ef1α-EGFP-WPRE-pA, mixed AAV2/8 serotype, 2.07E13 vg/mL) vectors. These were applied together at a 1:1 ratio and diluted 10 times in DPBS before use. For the RAM-ChR2 and RAM-ArchT experiments, we used homemade AAV-RAM-ChR2:EYFP (AAV-RAM-d2TTA-pA:: TRE-ChR2:EYFP-WPRE-pA, serotype 9, 4.33E13 vg/mL), AAV-RAM-ArchT:EGFP (AAV-RAM-d2TTA-pA::TRE-ArchT:EGFP-WPRE-pA, serotype 9, 2.19E13 vg/mL) and AAV-TRE-mCherry (AAV-TRE-mCherry-pA, serotype 9, 2.24E13 vg/mL). For a subset of luciferase experiments, we produced AAV viruses (mixed AAV2/8 serotype) of pAAV-TK-FLEX-Renilla-WPRE-pA (7.36E12 vg/mL) and pAAV-RAM-FLEX-luc2p-WPRE-pA (5.13E12 vg/mL). Both these constructs were cloned in our laboratory. For Cre-dependent RAM (CRAM) expression, we produced AAV-CRAM-tdTomato virus (AAV-RAM-d2TTA::TRE-FLEX-tdTomato-WPREpA, serotype 2/8, 1.22E13 vg/mL). Canine adenovirus serotype 2 (CAV2) (*Hnasko et al., 2006*) was engineered to express Cre recombinase, CAV2-Cre (10.3E12 pp/mL; kindly provided by Dr. Eric J. Kremer). Genomic AAV titer was determined by a PicoGreen-based method as described elsewhere (*Piedra et al., 2015*). Before use, all viruses were carefully examined in pilot experiments and, if needed, diluted in DPBS for optimized titer.

## Mouse subjects

Adult male C57Bl/6 mice (Charles River Laboratory) were 7–10 weeks of age for all surgeries and behavioral manipulations. Gad2-Cre (*Gad2$^{tm2(cre)Zjh}$*/J, Jackson Laboratory) mice were used for CRAM experiments. Following stereotaxic viral vector injections mice were single-housed for 7–14 days. All mice were housed in a temperature-controlled facility with a 12 hr light/dark schedule and provided with food and water *ad libitum*. All mouse protocols were performed in accordance with NIH guidelines and approved by the Massachusetts Institute of Technology Committee on Animal Care (protocol #1014-105-14).

## Rat subjects

Adult male Sprague-Dawley rats (275–300 g; Harlan, Indianapolis) were pair housed in a temperature- and humidity-controlled room on a 12 hr light/dark cycle (lights on at 7:00 A.M.). Standard lab chow and water were available *ad libitum*. Rats were allowed to acclimate to colony conditions for 7–10 days prior to surgery. All rat procedures were performed in accordance with NIH standard ethical guidelines and were approved by the Institutional Animal Care and Use Committee at the University of Colorado Boulder (protocol #1505.07).

## Viral vector delivery

Mice were placed on doxycycline chow 24 hr prior to surgery (40 mg/kg, Bio-Serv, Flemington, NJ). On the day of surgery, mice were anaesthetized using isoflurane (3% induction, 1.5% maintenance during surgery) and secured to a stereotactic frame (Kopf) with a heating pad to maintain body temperature. Mice were injected sub-dermally with Buprenex (1 mg/kg) and given topical Lidocaine (20 mg/mL) for analgesia (Hi-Tech Pharmacal, Amityville, NY). Following exposure of the skull, a small craniotomy was made overlying the target brain structure. Using a glass pipette (50 µm tip) connected to a Nanoject II (Drummond Scientific, Broomall, PA) injector, AAVs were delivered (50 nL/infusion, 30 sec between infusions) to the brain, and allowed to diffuse for 10 min before

withdrawing the pipette. The coordinates of the target brain structures in reference to bregma (mm) and injection volumes (nL) were as follows: dentate gyrus (AP: −1.9, ML: ± 1.4, DV: −2.05; 600), dorsal CA3 (AP: −1.9, ML: ± 1.85, DV: −2.1; 500), basolateral amygdala (AP: −1.6, ML: ± 3.25, DV: −3.6 from pial surface; 1000), and central amygdala (AP: −1.4, ML: ± 2.85, DV: −3.7 from pial surface; 300). All injections were performed bilaterally. Mice were kept on doxycycline chow and allowed to recover for 7–14 days following surgery.

Rats were anesthetized with isoflurane. A midline incision was made to expose the skull, and a small craniotomy was made unilaterally above the medial prefrontal cortex (mPFC). Recombinant AAV was injected into the prelimbic cortex (PL) of the mPFC. A stainless steel needle with the beveled tip facing laterally (31 gauge; Hamilton Company, Reno, Nevada) was directed to the PL (AP: +2.5, ML: ±0.5, DV: −2.0 from pial surface). For the CRAM experiment in rats, CAV2-Cre was bilaterally injected to the dorsal medial striatum (AP: +0.1, ML: ± 2.0, DV: −3.5 from pial surface). Virus (1.0 µL/hemisphere) was infused over 10 min (0.1 µL/min), followed by an additional 10 min to allow diffusion of the virus from the needle tip. The skin was sealed with Vetbond tissue adhesive.

## Rodent behavior

Doxycycline chow was typically withdrawn 48 hr prior to behavioral manipulation and replaced with regular feed, but see figure legends for precise experimental setup.

## Contextual fear conditioning

To assess behavior-specific induction of RAM in the hippocampus, we trained mice in a context-dependent classical conditioning paradigm following our previously established protocol (*Ramamoorthi et al., 2011*). Briefly, mice were placed in a box with salient visual cues, and allowed to explore for 58 s terminated by a 2 s, 0.55 mA foot-shock. This was repeated three times at 58 s intervals. Mice were kept in the chamber for an additional 60 s before being placed back into their home cages. Mice experiencing the conditional stimuli only (Context Only) were left for 240 s in the box, whereas mice experiencing the unconditional (Shock Only) stimuli only received the foot-shocks immediately once placed in the chamber and were then returned back to their home cage.

## Tone fear conditioning

To examine behavior-specific induction of RAM in the amygdala, we utilized an associative tone-fear conditioning paradigm, once again following an established protocol (*Ramamoorthi et al., 2011*). Briefly, mice were placed in a chamber and allowed to acclimate for 40 s. A tone (2.5 kHz, 85 dB, 20 sec) co-terminating with a foot-shock (0.55 mA, 2 s) was presented four times at 40 s intervals. The mice were given an additional 60 s in the chamber following the final tone presentation before being placed back into their home cages. A subset of mice received the conditional (Tone) stimuli only, but was otherwise handled similarly. Mice experiencing the unconditional (Shock) stimuli only received the foot-shocks immediately once placed in the chamber and were then returned back to their home cage.

## IEG overlap experiment

In order to characterize the extent of IEG protein expression in the same brain sections as RAM expressing cells, animals were run through the contextual fear conditioning paradigm (context A) listed above. The following day, mice were placed either back into the box from the original conditioning (context A) or into a novel context (context B) for 4 min and manually scored for freezing behavior every 5 s. 1.5 hr after the recall testing, animals were sacrificed and their brains were processed for immunohistochemical analysis.

## Drug induced seizures

Pharmacologically induced seizures were used to drive maximal RAM or IEG expression in the hippocampus or cortex. Mice were given intraperitoneal injections of 15 mg/kg kainic acid (Sigma), pH 7.4, or 50 mg/kg pentylenetetrazole (Sigma), and were selected for further analysis only if they exhibited full motor seizures.

## Inescapable stress

Rats were exposed to a single session of inescapable stress (IS) in Plexiglas boxes (14 × 11 × 17 cm) with a Plexiglas rod protruding from the rear. The rat's tail was secured to the rod with tape and affixed with copper electrodes. Each session consisted of 100 trials of tail shock (5 sec duration, 33 × 1.0 mA, 33 × 1.3 mA, and 34 × 1.6 mA) on a random-interval 1 min schedule. Subjects were returned to the colony immediately following the tail shock procedure.

## Immunohistochemistry

All histological analysis of RAM expression was performed 24 hr following relevant stimulus presentation. Mice and rats were deeply anesthetized with isoflourane and the brains were removed and drop-fixed in 4% paraformaldehyde in PBS. Brains were fixed for 24 hr at 4°C and then cryoprotected in 30% sucrose in PBS at 4°C until they sank. Subsequently, brains were sectioned on a cryostat at 50 μm thickness. All sections were blocked for 2 hr at room temperature in a TBS (pH 8.0) solution containing 0.5% tritonX-100, 0.2% BSA, 10% normal goat serum and then incubated with primary antibody in blocking solution, typically overnight at 4°C. The next day, sections were rinsed in PBS and incubated in secondary antibody for 2 hr at room temperature. Sections were mounted on Superfrost slides and coverslipped in DAPI Flouromount-G (SouthernBiotech). Primary antibodies used: FOS (rabbit, 1:1000, Santa Cruz sc-52), NPAS4 (rabbit, 1:10,000) (*Lin et al., 2008*), Glutamate Receptor 2&3 (GLUR2/3, rabbit, 1:200, Millipore AB1506), Somatostatin (SST, rat, 1:200, Millipore MAB354), Parvalbumin (PV, mouse, 1:1000, Millipore MAB1572), α-mKate2 (tRFP antibody recognizing mKate2, rabbit, 1:200, Evrogen AB223) and GAD67 (mouse, 1:1000 or 1:500 Millipore MAB5406). Secondary antibodies used: Alexa Fluor 405 (1:500, Thermo Fisher) and Alexa Fluor 647 (goat anti-rabbit IgG, goat anti-rat IgG, or goat anti-mouse IgG, 1:1000, Life Technologies).

## Image acquisition and cell quantification

For mouse brains, low magnification images were acquired from coronal sections using an Olympus BX51 fluorescent microscope using a 10X or 20X objective and MetaMorph software (Molecular Devices, Sunnyvale, CA). High magnification images were acquired from coronal sections using an Olympus Fluoview FV1000 with a 60X oil-immersion objective and Fluoview imaging software. All images were pseudo colored, merged and quantified using ImageJ software. Cell counts were performed using the Cell Counter plug-in. Briefly, all infected cells were confirmed by using the DAPI channel, and individual counts were taken for EGFP and RAM positive cells in order to provide a percentage of RAM+ cells among the infected population. Images were only taken in regions of maximum viral infectivity as determined by a near 100% neuronal expression of the EGFP infection marker. Counts were quantified from 3–5 separate sections from each injected hemisphere from 3–9 animals per condition.

For rat tissue, image acquisition was performed with a confocal system (ZeissLSM510; Carl Zeiss) using 5X and 20X objectives. Confocal images were acquired using identical pinhole, gain and laser settings. For each subject, the number of fluorescence mKate2 and tdT positive cells were quantified and averaged from 2–4 tissue sections of the PL.

## IEG overlap experiment imaging and analysis

Mice were exposed to a contextual fear conditioning paradigm and were then placed back into their home cages to allow for hippocampal RAM expression. Twenty-four hours later, the mice were divided into two cohorts and were either re-exposed to the initial conditioning context, or exposed to a novel context, and were then sacrificed 1.5 hr later. Alternating 50 μm brain slices were stained for NPAS4 or FOS. Coronal sections were imaged using an Olympus Fluoview FV1000 confocal microscope with a 60X oil objective and Fluoview imaging software. Images were only taken in regions of maximum viral infectivity as mentioned previously. For each animal (n = 4–5 per condition), 7–10 images were taken of the granule cell (GC) layer of the dorsal dentate gyrus, focusing on the apex of stratum granulosum. Each image was acquired as a Z-stack 20–35 μm thick, with a step size of 2.5 μm (slice thickness = 0.884 μm). After acquisition, RAM+, IEG+ and RAM+/IEG+ cells were confirmed to be DAPI positive and were manually counted for each stack in Fluoview. Overlap was defined as the percentage of RAM+ cells that were also IEG+. We estimated the total number of GCs per image as follows. High-resolution cubes (n = 18) of 35 × 35 × 10 μm were imaged in the

DAPI channel with a 1 µm step size. We estimated the diameter of each DAPI labeled GC nuclei to be 7.5 µm. For the entire cube, DAPI cells were counted, excluding those touching 3 of the 6 image faces, and the count was divided by the total cube volume to get a GC density estimate. Our overall GC density was calculated to be $1.35 \pm 0.06$ cells/1000 µm$^3$, slightly higher than previously published estimates (*Deng et al., 2013*; *Kempermann et al., 1998*). For each slice, total DG granule cell layer area was measured in Fluoview and multiplied by the stack thickness to get a GC volume. The calculated GC density was then used to give a total GC estimate per slice.

## Whole-cell patch-clamp recordings of RAM-ChR2 and RAM-ArchT cells in acute slice preparations

Mice were decapitated 24 hr following CFC. The brains were removed from the skulls and immediately immersed in carbogenated ice-cold cutting solution containing (in mM) sucrose 75, NaCl 67, NaHCO$_3$ 26, glucose 25, KCl 2.5, NaH$_2$PO$_4$ 1.25, CaCl$_2$ 0.5, MgCl$_2$ 7, 310 mOsm, pH 7.35. In the same solution, 300 µm horizontal slices were cut with a vibrating blade microtome (VT1200, Leica, Germany). Slices containing the dorsal hippocampus were transferred to an incubation chamber filled with carbogenated warm (32˚C) cutting solution for 10–20 min and then at room temperature (~23˚C) for at least 1 hr before being used. For recording, slices were transferred to a recording chamber perfused at a flow rate of 2 ml/min with artificial cerebrospinal fluid (ACSF) containing (in mM) 119 NaCl, 2.5 KCl, 1.24 NaH$_2$PO$_4$, 1.3 MgCl$_2$, 2.5 CaCl$_2$, 26 NaHCO$_3$, and 10 glucose, carbogenated with 95% O$_2$ and 5% CO$_2$ gas mixture and kept at room temperature. Borosilicate glass pipettes (4–6 MΩ tip resistance) were used for whole-cell patch-clamp recordings and filled with internal solution as follows (in mM): 130 K-gluconate, 10 KCl, 1 MgCl$_2$, 10 HEPES, 0.2 EGTA, 4 Mg-ATP, 0.5 Na-GTP, pH 7.25, 290 mOsm osmolarity.

RAM+ neurons expressing ChR2 or ArchT were recorded in a mixture of 50 µM picrotoxin, 50 µM APV and 20 µM DNQX to block fast GABAergic and glutamatergic synaptic transmission. A square pulse of −5 mV was delivered at the end of every sweep to monitor access resistance. Recordings with access resistance greater than 25 MΩ or with changes in access resistance greater than 15% were discarded. ChR2 and ArchT expressing RAM+ cells were illuminated with 455 nm blue and 565 nm green light, respectively, through a 40X optical lens, and the light sources (LEDs, Thorlabs, Newton, NJ) were controlled by the pClamp10 interface. Data were collected with a Multiclamp 700 b amplifier (Molecular Devices), Bessel filtered at 2 kHz. Data were digitized at 10 kHz using a Digidata 1440A (Molecular Devices) and were acquired using pClamp10 software. Photocurrents were measured using pClamp10.

## Generation of RAM reporter fly

Three RAM binding sites were placed upstream of a minimal promoter and CaSpeR TATAA sequence, followed by flippase recognition target (FRT)-flanked stop codons and the luciferase open reading frame. A short sequence coding for a poly-glycine run was placed downstream of the second FRT site and arranged so that it was in the same reading frame as the ATG start codon, regardless of which FRT site remained after site-specific recombination. The luciferase-coding region (minus its normal ATG start codon) was placed downstream and in frame with the poly-glycine run. In the absence of the flippase (FLP), the transgene would produce no luciferase protein. After FLP-mediated recombination, a fusion protein would be expressed that contains amino acids from the FRT sequence and a poly-glycine run, all fused to luciferase. Reporter constructs were inserted at the NotI/XhoI sites of pCaSper5. Standard methods were used to generate transgenic flies (BestGene). The specificity of the RAM-luc to Flp recombinase was tested using a UAS-Gal4 system, in which Flp recombinase expression is under the control of a heat-shock (hs) promoter (see *Figure 6e–f*).

## Fly behavioral training

Flies were trained in the olfactory avoidance-training paradigm developed by Tully and Quinn and modified to allow for automated training sessions (*Fropf et al., 2013*). A single cycle of training consists of 90 s exposure to ambient air; 60 s of electric shock (the unconditioned stimulus); 70 olt pulses lasting 1.5 s and administered every 5 s (12 total) accompanied by simultaneous exposure to 1 odor (the conditioned stimulus condition, CS+); 45 sec of ambient air exposure to clear the first odor; 60 s

of exposure to the second odor, with no shock (the CS- condition), 45 s of ambient air to clear the second odor. Spaced training consists of ten single cycles separated by 15 min rest intervals. We used 3-octanol and 4-methylcyclohexanol as odors. One group of flies was exposed to Forward Spaced (FS) training, where the odor was paired with shock presentation. The control group of files was exposed to the same number of training trials, but the non-overlapping odor and shock presentations did not overlap in time (Backward Spaced, or BS training), which does not produce associative memory.

### In vivo luciferase assays in flies

Flies were housed at 50 flies per vial and entrained on a 12 hr light/dark cycle for 3–5 days before beginning experiments. 24 to 48 flies were loaded into a black 96-well microplate containing luciferin media: 1% agar, 5% sucrose, 5 mM D-Luciferin (Gold BioTechnology, Olivette, MO). Plates were maintained at 22°C under controlled light conditions and cycled approximately once per hour through a Packard TopCount Scintillation and Luminescence counter. To display oscillatory activity of pan-neuronal RAM-luc reporter (*Figure 6g*), a smoothing function was applied such that each data point represented the average of three measurements. To compare overall activity between groups (*Figure 6h*), the mean hourly reading was calculated for each fly and compared.

### Statistics

All statistical information is provided in the figure legends and *Supplementary file 2*. Data obtained from the 10mer screen were analyzed using a binomial test (see 'Screening for Enriched Enhancer' for details). All other data were compiled and analyzed using GraphPad Prism 5.0 software. Unpaired Student's t-test, one-way ANOVA with Tukey or Dunnet post-hoc test, and two-way ANOVA with Bonferroni post-hoc test were used whenever appropriate. The level of significance was set at $p < 0.05$. All data are presented as mean ± SEM. No estimates of power were performed before experiments, but sample size numbers were similar to those generally employed in the field.

## Acknowledgements

We thank C. Mark Fletcher, Charles Jennings and Catherine Dulac for critical reading of the manuscript, Michael Cooper for editing the manuscript, Haruhiko Bito for kindly providing the ESARE plasmid, and Jonathan Ting for sharing protocols for AAV production. We are grateful to Danica Rili for assisting with dissections and primary neuron cultures and Xiaomeng Han for assisting with some luciferase experiments. This work was funded by a Swedish Brain Foundation Research Fellowship (ATS), a MIT Simons Initiative on Autism and the Brain Postdoctoral Fellowship (AY), a NRSA pre-doctoral fellowship (KR), NIH Training Grant T32 GM007507 and NSF Graduate Research Fellowship DGE-1256259 (RF), NIH Grant MH093412 (SFM), NIH Grant MH106817 (MVB), NIH grants MH067774 and NS063245 (JY), Wellcome Trust (MH) and the John Merck Scholar Program and NIH grants MH091220 and NS088421 (YL).

## Additional information

### Funding

| Funder | Grant reference number | Author |
|---|---|---|
| Hjärnfonden | Research Fellowship | Andreas Toft Sørensen |
| National Institutes of Health | MH106817 | Michael V. Baratta |
| National Institutes of Health | NRSA Pre-doctoral Fellowship | Kartik Ramamoorthi |
| National Institutes of Health | T32 GM007507 | Robin Fropf |
| National Science Foundation | Graduate Research Fellowship DGE-1256259 | Robin Fropf |
| Simons Foundation | MIT Simons Initiatives on Autism and the Brain, Postdoctoral Fellowship | Andrew Young |

| | | |
|---|---|---|
| Wellcome Trust | | Martin Hemberg |
| National Institutes of Health | MH067774 | Jerry CP Yin |
| National Institutes of Health | NS063245 | Jerry CP Yin |
| National Institutes of Health | MH093412 | Steven F Maier |
| John Merck Fund | Scholars Program | Yingxi Lin |
| National Institutes of Health | MH091220 | Yingxi Lin |
| National Institutes of Health | NS088421 | Yingxi Lin |

The funders had no role in study design, data collection and interpretation, or the decision to submit the work for publication.

### Author contributions

ATS, cloned and produced all plasmids and AAV constructs, screened and characterized all activity reporters in vitro, conducted characterization of RAM in vivo, conducted all electrophysiology experiments, was a main contributor to all data analysis and interpretation, conceived the project, designed experiments, wrote the manuscript with inputs from all other co-authors; YAC, cloned and produced all plasmids and AAV constructs, conducted characterization of RAM in vivo, was a main contributor to all data analysis and interpretation, designed experiments, wrote the manuscript with inputs from all other co-authors; MVB, SFM, designed and conducted all rat experiments; F-JW, conducted characterization of RAM in vivo, conducted all electrophysiology experiments; YZ, JX, conducted characterization of RAM in vivo; KR, cloned and produced all plasmids and AAV constructs, contributed to the initial characterization of RAM, conceived the project; RF, JCPY, designed and conducted all fly experiments; EL, cloned and produced all plasmids and AAV constructs, screened and characterized all activity reporters in vitro; AY, CS, CRG, contributed to the initial characterization of RAM; MH, conducted bioinformatics analysis; YL, conceived the project, designed experiments, wrote the manuscript with inputs from all other co-authors

### Author ORCIDs

Yingxi Lin, http://orcid.org/0000-0002-7002-1275

### Ethics

Animal experimentation: All mouse protocols were performed in accordance with NIH guidelines and approved by the Massachusetts Institute of Technology Committee (Protocol #1014-105-14). All rat procedures were performed in accordance with NIH standard ethical guidelines and were approved by the Institutional Animal Care and Use Committee at the University of Colorado Boulder (Protocol #1505.07).

## Additional files

### Supplementary files

• Supplementary file 1. A full list of AP-1 containing 10mers (128 in total) ranked by their enrichment factor. See 'Methods' for further details. 10mers used in $P_{RAM}$ and enhancer elements E1-E3 (*Figure 1a*, and *Figure 1—figure supplement 2b*) are highlighted in bold. 10mers with corrected p-value$\geq$0.05 are marked in grey color.

• Supplementary file 2. Experimental conditions and statistics.

### Major datasets

The following previously published dataset was used:

| Author(s) | Year | Dataset title | Dataset URL | Database, license, and accessibility information |
|---|---|---|---|---|
| Kim TK, Hemberg M, Gray JM, Costa AM, Bear DM, Wu J, Harmin DA, Laptewicz M, Barbara-Haley K, Kuersten S, Markenscoff-Papadimitriou E, Kuhl D, Bito H, Worley PF, Kreiman G, Greenberg ME | 2010 | Widespread transcription at neuronal activity-regulated enhancers | http://www.ncbi.nlm.nih.gov/geo/query/acc.cgi?acc=GSE21161 | Publicly available at the NCBI Gene Expression Omnibus (accession no. GSE21161) |

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
