## [Decision Letter]

Thank you for submitting your manuscript "A robust activity marking (RAM) system for exploring neuronal activity" as a Tools and Resources article to *eLife*. Three experts reviewed your manuscript, and the evaluation was overseen by Jeremy Nathans as the Reviewing Editor and a Senior Editor. Their assessments, together with my own, form the basis of this letter. As you will see, all of the reviewers were impressed with the importance and novelty of your work.

I am including the three reviews (lightly edited) at the end of this letter, as there are a variety of specific and useful suggestions in them. One point, in particular, that we noted is the limited overlap of P_RAM_ with Fos or *Npas4*. This needs to be explained.

As you will see, all of the reviewers were impressed with the importance and novelty of your work.

*Reviewer #1:*

Manuscript 13918 by Sørensen et al. describes a new method to identify active neurons in vivo and to conditionally express transgenes. The method appears robust and adaptable to many uses. In my opinion the work presents an important advance and is suitable for publication in *eLife*.

The manuscript begins with a generalizable method to identify candidate cis elements and focuses on binding sites for *Npas4* and AP-1. They go on to demonstrate the robust inducibility and low basal in depolarized cultured neurons and then in vivo in response to context fear conditioning and kainic acid induced seizure. This looks to be superior to previous methods and so represents an important advance. I was surprised that there is a relatively small difference in Figure 4 between A-A and A-B. This may make people worry that it is not really identifying behaviorally linked networks. I wonder why the authors did not look at CA1 neurons in a place field paradigm where the percentage of active neurons and network specificity is well established. Might the threshold activity required to induce the system be relatively high compared to Arc for example. If so, this should be mentioned in the manuscript. The strategy to confer Cre dependence presented in Figure 5 is very nice. Did you ever see expression in parvalbumin neurons? Comment?

I would have liked to see a panel in Figure 2 looking at dentate 48 hrs after removal of Dox but without kainic acid.

Figure 4 could benefit from a schematic showing the injection site.

Data in Figure 3—figure supplement 3 is very nice showing optogenetic control and might be substituted for panels E-G.

The statistical analysis and data provided appear appropriate.

*Reviewer #2:*

Sørensen and colleagues have built a synthetic promoter (P_RAM_) toolkit that reports neuronal activation. Such a promoter can, in principle, be utilized in different ways to mark and manipulate a target neuronal population. The authors provide two such examples: incorporating P_RAM_ with a destabilized tTA in a tet-off system into a single AAV vector that can be used in vertebrate brains and using P_RAM_ to drive reporter expression in a Gal4 and Flp dependent manner in *D. melanogaster*. I think this system is likely to be widely used, and I am supportive of publication in *eLife*. In general the experiments are elegantly designed and executed, but I have a few concerns relating to the sensitivity and specificity of labeling of active neurons.

1) The authors make strong claims about the superior sensitivity of their system compared to previous approaches. These may well be true, but it would nice to know how sensitive their system is in terms of detecting changes in neuronal activation. For example, what is the minimal percent increase in firing from baseline that is reliably reported by P_RAM_ ? A measure of sensitivity would also provide estimates of how much noisy fluctuations in firing will elicit P_RAM_ driven transcriptional activity. At the very least this should be discussed. The authors could also potentially test this in their primary neuronal cultures (Figure 1), ideally with channelrhodopsin mediated control of firing. Alternatively, rather than just testing P_RAM_ activity with 35mM KCl for 6 hours, a time course (say 2 and 4 hours) and a dose response to KCl would be informative. In these studies, it would be valuable to perform immunolabeling for c-Fos in parallel to compare this new approach to one that is familiar to most neuroscientists.

2) Some controls appear to be missing for the in vivo studies relating to kainic acid induced seizures (Figure 2) and contextual or tone conditioning (Figure 3, Figure 5). How many neurons express the reporter 24 hours following:saline rather than kainic acid injection;placement in the contextual testing chamber without being shocked;hearing the tone in the testing chamber without being shocked?

These controls will reveal how likely the neurons under consideration express reporter without the experimental manipulation being tested.

3) I really appreciate the authors' extending their validation of reporter induction to multiple brain regions and Glu and Gabaergic neurons. However, the number of reporter+ neurons is comparable between home cage and contextual conditioning paradigms (Figure 5—figure supplement 1A-C), so it's unclear whether reporter expression in Gabaergic Sst+ neurons (panel d) represents reporter induction in response to conditioning or basal expression seen in control home cage conditions.

As far as I can tell, there is no control for the aggressive social interaction test of reporter induction in the prefrontal cortex (Figure 5—figure supplement 1G-H). What is the level of basal reporter expression in the home cage in prefrontal cortex? A comparison between the control and aggression conditions would reveal whether there is reporter induction in the Gabaergic neurons in this region.

1) Please provide a schematic of the experimental setup for the in vivo CRAM studies in Figure 5.

2) Clarify what "% overlap" means as the y-axis label in Figure 4.

3) The first paragraph of the subsection “Application of the RAM System to Other Species” needs to be corrected; AAV-RAM was tested in rats and P_RAM_ was tested in flies.

4) In the second paragraph of the subsection “Application of the RAM System to Other Species”: cite study showing PFC projection to dorsal striatum.

5) Correct discrepancy between Methods and Supplementary Figure legend: was the social interaction/aggression test done for 5 or 10 min?

6) The authors will be depositing the basic AAV-RAM vector with Addgene. I also encourage them to deposit the CRAM version of the vector and other plasmids described here, including the ones for flies (but see below).

7) Frankly, the fly data seem a bit rushed. For example, do we even know whether luciferase expression is even restricted to the nervous system? I would suggest that the authors hold on to the fly data and submit it as a more complete, separate story.

*Reviewer #3:*

This manuscript by Sørensen et al. reports extensive characterization of a new tool for marking and manipulating neurons based on immediate early gene expression. This tool has the potential to be an important new addition to the arsenal for defining and manipulating cell populations based on neuron activity changes. They developed the RAM promoter, which gives improved signal to noise over past IEG systems and also is compact enough to fit easily inside an AAV vector. This promoter can be used to drive fluorescent proteins and optogenetic neuronal actuators. They go on to demonstrate the application of this tool in mice in a seizure model, contextual fear conditioning, auditory conditioning as well as other models in rat and fly. In addition, a Cre-dependent variant on the tool is described.

The tool depends on the tetOff tTA system, and doxycycline is administered to reduce background expression after viral expression and prior to the experiment. However, there is some ambiguity here because in the rat model, the dox was excluded and low background was apparently maintained. The use of dox appears to be important for the use of this tool, but the necessity of this component needs to be more explicitly expanded upon so that a future user is well-positioned to understand its importance. In addition, further explanation is required for the modest colocalization of Fos (note Fos, not c-Fos, is the MGI protein abbreviation) with mKate2 in Figure 4 for the A-A conditions (only about 20%). Although this colocalization is significantly greater than the context A-B, there is no explanation why the colocalization is not higher. Because the expectation is that expression off of this promoter reflects IEG expression in neurons, this discrepancy is especially concerning as it is an essential aspect for application of this tool. Further examination or explanation of this issue is required.

---

## [Author Response]

*Reviewer #1:*

*Manuscript 13918 by Sorensen et al. describes a new method to identify active neurons in vivo and to conditionally express transgenes. The method appears robust and adaptable to many uses. In my opinion the work presents an important advance and is suitable for publication in eLife.*

*The manuscript begins with a generalizable method to identify candidate cis elements and focuses on binding sites for Npas4 and AP1. They go on to demonstrate the robust inducibility and low basal in depolarized cultured neurons and then in vivo in response to context fear conditioning and kainic acid induced seizure. This looks to be superior to previous methods and so represents an important advance. I was surprised that there is a relatively small difference in Figure 4 between A-A and A-B. This may make people worry that it is not really identifying behaviorally linked networks. I wonder why the authors did not look at CA1 neurons in a place field paradigm where the percentage of active neurons and network specificity is well established. Might the threshold activity required to induce the system be relatively high compared to Arc for example. If so, this should be mentioned in the manuscript. The strategy to confer Cre dependence presented in Figure 5 is very nice. Did you ever see expression in parvalbumin neurons? Comment?*

The reviewer made the very insightful point that the CA1 region is much better established in a place field paradigm. We chose to focus on the dentate gyrus (DG) in this study mainly out of experimental convenience. The DG has been extensively examined in contextual fear conditioning (CFC), making it an excellent region to validate activity reporters. By targeting DG, not only can our results be conveniently compared to other published studies, it also allows us to examine the specificity of the RAM system using the A-A/A-B type of experiments (Figure 4). Being an important region for place field paradigm, CA1 should be thoroughly examined in the future.

There are two concerns raised about The A-A/A-B experiment (the original Figure 4). First, the low percentage (~20%) of overlap between RAM+ and Fos+ neurons in the A-A experiment (Reviewer #3). Second, the relatively small difference in overlap between A-A (~22%) and A-B (~13%) conditions (Reviewer #1).

These are legitimate concerns. To address these concerns, we have carefully examined published studies. Since the behavioral manipulations involved vary considerably, the IEGs used for these experiments are not the same, and the quantification methods and statistical analyses also vary significantly, it is extremely difficult to directly compare numbers between studies. Nevertheless, we tried our best to compare our data to studies done in dentate gyrus using contextual fear conditioning. We found our results to be either comparable to or even more impressive than published studies. For example, a study from the Gage lab (Figure 4 in Deng et al. *eLife* 2013;2:e00312) reported a ~6% overlap between Fos reporter (Fos-tTA::*tetO-tau-lacZ:tTA**) positive and Fos+ neurons for A-A exposure. Another study from Dr. Rene Hen’s lab (Figure 3 in Denny et al. Neuron 83, 2014) showed a 5% or 4% overlap between the Arc-CreER reporter positive neurons and Fos+ or Arc+ neurons for A-A exposure, respectively. While the Gage study didn’t detect a difference in overlapping population between A-A and A-B, the Hen study showed a significant difference between A-A (~4%) and A-B (~2%) conditions. Our data are perhaps best aligned with the study from the Tonegawa lab (Figure 2 in Ramirez et al. Science 341, 2013), in which they reported, using a very different quantification method, ~1.8% vs. ~0.9% DAPI+ cells being doubly labeled by the Fos-tTA reporter and Fos antibody under A-A vs. A-B conditions. Assuming the Fos-tTA reporter labeled ~6% neurons following CFC (based on another study from the Tonegawa lab, Figure 2 in Liu et al. Nature 484, 2012), for the Ramirez study we can estimate that ~30% and 15% of cells labeled by Fos-tTA were also Fos+ for A-A and A-B conditions respectively. This is comparable to our study. Critically, although the overall number of overlapping neurons varied wildly between studies, the trend is a roughly 2-fold increase in overlap between A-A and A-B conditions. In this respect, our data falls in line with the published literature. We have emphasized this point in the revised manuscript.

*I would have liked to see a panel in Figure 2 looking at dentate 48 hrs after removal of Dox but without kainic acid.*

This is basically the home cage control in Figure 3 and D.

*Figure 4 could benefit from a schematic showing the injection site.*

A schematic showing the injection site has been included in the new Figure 4.

*Data in Figure 3—figure supplement 3 is very nice showing optogenetic control and might be substituted for panels E-G.*

The optogenetic control is kept as supplementary data. Instead the Context Only condition has been added to Figure 3.

*Reviewer #2:*

*Sørensen and colleagues have built a synthetic promoter (P_RAM_) toolkit that reports neuronal activation. Such a promoter can, in principle, be utilized in different ways to mark and manipulate a target neuronal population. The authors provide two such examples: incorporating P_RAM_ with a destabilized tTA in a tet-off system into a single AAV vector that can be used in vertebrate brains and using P_RAM_ to drive reporter expression in a Gal4 and Flp dependent manner in D. melanogaster. I think this system is likely to be widely used, and I am supportive of publication in eLife. In general the experiments are elegantly designed and executed, but I have a few concerns relating to the sensitivity and specificity of labeling of active neurons.*

*1) The authors make strong claims about the superior sensitivity of their system compared to previous approaches. These may well be true, but it would nice to know how sensitive their system is in terms of detecting changes in neuronal activation. For example, what is the minimal percent increase in firing from baseline that is reliably reported by* P_RAM_*? A measure of sensitivity would also provide estimates of how much noisy fluctuations in firing will elicit* P_RAM_
*driven transcriptional activity. At the very least this should be discussed. The authors could also potentially test this in their primary neuronal cultures (Figure 1), ideally with channelrhodopsin mediated control of firing. Alternatively, rather than just testing* P_RAM_
*activity with 35mM KCl for 6 hours, a time course (say 2 and 4 hours) and a dose response to KCl would be informative. In these studies, it would be valuable to perform immunolabeling for c-Fos in parallel to compare this new approach to one that is familiar to most neuroscientists.*

This is an excellent point. To the best of my knowledge, no one has systematically examined the activation sensitivity of any of the activity reporters being used in the field. Systematic characterization of activity reporters is very important for the neuroscience field and we are actively working on this using both in vitro and in vivo approaches. However, the process is quite involved. Most of the approaches we tried in vitro in cultured neurons, including using channelrhodopsin and KCl dosage as suggested by Reviewer #2, turn out to be quite invasive and non-physiological. We are currently developing better methods to tackle this problem. As such, we don’t have results to answer the question, but we have included this very important point in the Discussion.

*2) Some controls appear to be missing for the in vivo studies relating to kainic acid induced seizures (Figure 2) and contextual or tone conditioning (Figure 3, Figure 5). How many neurons express the reporter 24 hours following:saline rather than kainic acid injection;placement in the contextual testing chamber without being shocked;hearing the tone in the testing chamber without being shocked?*

*These controls will reveal how likely the neurons under consideration express reporter without the experimental manipulation being tested.*

There are some points regarding the activity induction of interneurons.

1) Can RAM label activated PV+ neurons (Reviewer #1)?

2) Reviewer #2 pointed out that the control for the social defeat experiment (original Figure 5—figure supplement 1G, H) was missing, making it unclear whether there is behaviorally-mediated reporter induction in the GABAergic neurons in the prefrontal cortex.

To address these concerns, we used inescapable stress paradigm to determine whether RAM labeling is induced in GABAergic neurons in the prefrontal cortex and if RAM can also label PV+ neurons. The answers to both questions are Yes. The result has been included in the revised manuscript, Figure 7 and Figure 7—figure supplement 1.

Reviewer #2 requested the following control experiments.

1) Saline injection control for kainic acid injection in the original Figure 3 and 5. The purpose of the seizure condition in Figure 3 and Figure 5 was to demonstrate the infection efficiency of the RAM virus, not to compare seizure vs. non-seizure condition. The saline injection control is therefore unnecessary.

2) Context without shock control in the original Figure 3.

We have included this control and, in addition, the immediate shock control in the revised manuscript (Figure 3). Consistent with published studies showing context alone significantly induces IEG expressions, the number of RAM+ neurons for context only is numerically very similar to the context + shock condition. Also, the immediate shock didn’t seem to activate RAM at all, which suggests neuronal ensembles labeled by RAM are specifically involved in contextual learning.

3) Amygdala TFC sound only control in the original Figure 5.

We have now included both tone only and shock only controls for the TFC experiment in the revised manuscript (Figure 5). In addition, we also included a comparison between LA and basal amygdala (BA). Our results are very much consistent with our current understanding of the functions of these two sub-regions of the amygdala, as well as IEG expression data for those regions after TFC. RAM only significantly labels LA under the tone + shock condition, but not by tone alone or shock alone. In contrast, all conditions significantly activated RAM in BA.

*3) I really appreciate the authors' extending their validation of reporter induction to multiple brain regions and Glu and Gabaergic neurons. However, the number of reporter+ neurons is comparable between home cage and contextual conditioning paradigms (Figure 5—figure supplement 1A-C), so it's unclear whether reporter expression in Gabaergic Sst+ neurons (panel d) represents reporter induction in response to conditioning or basal expression seen in control home cage conditions.*

*As far as I can tell, there is no control for the aggressive social interaction test of reporter induction in the prefrontal cortex (Figure 5—figure supplement 1G-H). What is the level of basal reporter expression in the home cage in prefrontal cortex? A comparison between the control and aggression conditions would reveal whether there is reporter induction in the Gabaergic neurons in this region.*

*1) Please provide a schematic of the experimental setup for the in vivo CRAM studies in Figure 5.*

*2) Clarify what "% overlap" means as the y-axis label in Figure 4.*

*3) The first paragraph of the subsection “Application of the RAM System to Other Species” needs to be corrected; AAV-RAM was tested in rats and* P_RAM_
*was tested in flies.*

*4) In the second paragraph of the subsection “Application of the RAM System to Other Species”: cite study showing PFC projection to dorsal striatum.*

The above points have been addressed in the revised manuscript.

*5) Correct discrepancy between Methods and Supplementary Figure legend: was the social interaction/ aggression test done for 5 or 10 min?*

The social interaction/aggression experiment in mice has been replaced with inescapable stress experiment in rats.

*6) The authors will be depositing the basic AAV-RAM vector with Addgene. I also encourage them to deposit the CRAM version of the vector and other plasmids described here, including the ones for flies (but see below).*

As stated in the revised manuscript, both AAV-RAM and AAV-CRAM plasmids, from which other plasmids used in the study can be derived, have been at Addgene and will be available upon the publication of our manuscript.

*7) Frankly, the fly data seem a bit rushed. For example, do we even know whether luciferase expression is even restricted to the nervous system? I would suggest that the authors hold on to the fly data and submit it as a more complete, separate story.*

Although I agree with the reviewer that the fly data are not as thorough as we might prefer, I feel that the RAM promoter will be of great interest to the fly community. As far as I know, there is no good activity reporter currently available for use in the fly. Upon discussing with you, the fly data is kept in the revised manuscript.

*Reviewer #3:*

*This manuscript by Sorenson et al. reports extensive characterization of a new tool for marking and manipulating neurons based on immediate early gene expression. This tool has the potential to be an important new addition to the arsenal for defining and manipulating cell populations based on neuron activity changes. They developed the RAM promoter, which gives improved signal to noise over past IEG systems and also is compact enough to fit easily inside an AAV vector. This promoter can be used to drive fluorescent proteins and optogenetic neuronal actuators. They go on to demonstrate the application of this tool in mice in a seizure model, contextual fear conditioning, auditory conditioning as well as other models in rat and fly. In addition, a Cre-dependent variant on the tool is described.*

*The tool depends on the tetOff tTA system, and doxycycline is administered to reduce background expression after viral expression and prior to the experiment. However, there is some ambiguity here because in the rat model, the dox was excluded and low background was apparently maintained. The use of dox appears to be important for the use of this tool, but the necessity of this component needs to be more explicitly expanded upon so that a future user is well-positioned to understand its importance. In addition, further explanation is required for the modest colocalization of Fos (note Fos, not c-Fos, is the MGI protein abbreviation) with mKate2 in Figure 4 for the A-A conditions (only about 20%). Although this colocalization is significantly greater than the context A-B, there is no explanation why the colocalization is not higher. Because the expectation is that expression off of this promoter reflects IEG expression in neurons, this discrepancy is especially concerning as it is an essential aspect for application of this tool. Further examination or explanation of this issue is required.*

Reviewer #3 pointed out that Dox was not used in experiments done in rats and wondered if the Dox control would work well in the rat. Although not included in the initial submission, we have also used Dox to control RAM expression in the rat, which yielded much more impressive labeling specificity. These experiments replace the old figure panels in Figure 6 in the revised manuscript.